# cGAS-STING are responsible for premature aging of telomerase-deficient zebrafish

Naz Şerifoğlu[1], Giulia Allavena [ID][1], Bruno Lopes-Bastos[1], Marta Marzullo [ID][2,6], Andreia Marques [ID][1,3], Pauline Colibert [ID][1], Pavlos Bousounis[4,5], Eirini Trompouki[1] & Miguel Godinho Ferreira [ID][1✉]

## Abstract

**Telomere shortening occurs in multiple tissues throughout aging. When telomeres become critically short, they trigger DNA-damage responses and p53 stabilization, leading to apoptosis or replicative senescence. In vitro, cells with short telomeres activate the cGAS-STING innate immune pathway resulting in type-I interferon-based inflammation and senescence. However, the consequences of these events for the organism are not yet understood. Here, we show that *sting* is responsible for premature aging of telomerase-deficient zebrafish. We generated *sting-/- tert-/-* double-mutant animals and observed a thorough rescue of *tert-/-* phenotypes. At the cellular level, lack of cGAS-STING in *tert* mutants resulted in reduced senescence, increased cell proliferation, and decreased inflammation despite similarly short telomeres. Critically, absence of *sting* function resulted in dampening of the DNA damage response and reduced p53 levels. At the organism level, *sting-/- tert-/-* zebrafish regained fertility, showed delayed cachexia, and decreased cancer incidence, resulting in increased healthspan and lifespan of telomerase mutant animals.**

**Keywords** Telomerase; cGAS-STING; Inflammation; Aging; Zebrafish
**Subject Categories** Immunology; Molecular Biology of Disease

## Introduction

Telomerase deficiency in humans results in the development of telomere biology disorders (TBDs) that include idiopathic pulmonary fibrosis, dyskeratosis congenita and aplastic anemia (Armanios, 2009). Common aspects of TBDs relate to accelerated telomere shortening, loss of tissue regeneration, premature aging phenotypes and shorter lifespan (Mitchell et al, 1999; Opresko and Shay, 2017). Zebrafish possess human-like telomere lengths that shorten to critical lengths during their lifetime (Ferreira, 2025). Like human TBDs, telomerase-deficient zebrafish (*tert-/-*) have accelerated telomere shortening, low cell proliferation, tissue damage, and reduced lifespan (Anchelin et al, 2013; Carneiro et al, 2016; Henriques et al, 2013). *tert-/-* zebrafish also develop chronic inflammation, increased infections, and accelerated incidence of cancer (Lex et al, 2020; El Maï et al, 2023). Like in humans (Oh et al, 2023), we have recently shown that not all organs age at the same rate (El Maï et al, 2023). The zebrafish intestine becomes dysfunctional earlier and triggers systemic aging. Reduced proliferation of intestinal cells results in loss of tissue integrity, microbiota dysbiosis and systemic inflammation (El Maï et al, 2023).

Type I interferon response and secretion of pro-inflammatory cytokines through activation of cGAS-STING is triggered by DNA damage, including telomeric damage, leading to the formation of micronuclei (MN) (Glück et al, 2017; Yang et al, 2017; Dou et al, 2017; Nassour et al, 2024). The cytosolic DNA sensor, cGAS (cGMP-AMP synthase), becomes activated upon binding to double-stranded DNA, encompassing both microbial and self-DNA (Motwani et al, 2019). Upon recognition of cytosolic DNA, cGAS initiates the production of the second messenger cGAMP that binds and activates the adapter protein STING (Sun et al, 2013; Ishikawa and Barber, 2008). Subsequently, STING recruits TBK1 (TANK-binding kinase 1), triggering the activation of IRF3 (IFN regulatory factor 3). This activation cascade leads to the generation of type I interferons and inflammatory cytokines (Motwani et al, 2019). cGAS-STING is involved in DNA repair, DNA damage responses (DDR) and cell senescence (reviewed in Zierhut, 2024). These effects are achieved through the expression of interferon-stimulated genes (ISGs) and the senescence-associated secretory phenotype (SASP) (Ishikawa and Barber, 2008; Dou et al, 2017). Secretion of these molecules modulates the proliferative capacity of surrounding cells in a paracrine manner, propagating the senescence status (paracrine-SASP) (Glück and Ablasser, 2019).

Type I interferon response is increasingly linked to aging and neurodegenerative diseases across different species. A recent significant study by the Ablasser lab using naturally aged mice revealed that cGAS–STING signaling plays a pivotal role in the age-related type I interferon response in neurodegeneration (Gulen et al, 2023). Transcriptional profiling revealed that cGAS–STING activation initiated a gene expression program shared between

[1]Institute for Research on Cancer and Aging of Nice (IRCAN), CNRS UMR7284, INSERM U1081, Université Cote d'Azur, 06107 Nice, France. [2]Instituto Gulbenkian de Ciência, Oeiras, Portugal. [3]Faculty of Medicine, University of Coimbra, Coimbra, Portugal. [4]Department of Cellular and Molecular Immunology, Max Planck Institute of Immunobiology and Epigenetics, Freiburg, Germany. [5]Faculty of Biology, University of Freiburg, Freiburg, Germany. [6]Present address: Department of Biology and Biotechnologies, Sapienza University of Rome, Rome, Italy. ✉E-mail: Miguel-Godinho.FERREIRA@unice.fr

neurodegenerative diseases and natural aging. The authors also observed that there was an accumulation of mitochondrial DNA in the cell cytoplasm of microglial cells, providing a potential mechanism through which the cGAS-STING pathway may contribute to inflammation in the aging brain (Gulen et al, 2023).

In vitro studies using human primary cells recently showed that cGAS-STING is activated by short or dysfunctional telomeres (reviewed in Nassour et al, 2024). However, it is currently unknown what are the consequences of activation of cGAS-STING in response to short telomeres at the organism level. Here, we show that telomere shortening triggers the cGAS-STING pathway in skin, testis, kidney marrow (the adult hematopoietic organ in fish), and intestine of zebrafish. This results in type I interferon response, elevation in senescence levels and reduction in proliferative capacity of these tissues. Absence of cGAS-STING restores cell proliferation, suppresses accelerated aging and increases in both health and lifespan of the telomerase-deficient zebrafish.

# Results

## Telomere shortening activates the cGAS-STING and type I interferon in vivo

Our previous studies show that telomerase-deficient zebrafish undergo accelerated systemic inflammation (Henriques et al, 2013; Carneiro et al, 2016; El Maï et al, 2023; Lex et al, 2020; Carneiro Madalena, 2015; El Maï et al, 2020). Analysis of gene expression of proliferative (gut) and non-proliferative tissues (muscle) derived from aged *tert-/-* (9 months old) and WT (36 months old) zebrafish highlighted several genes related to type I interferon response (Carneiro Madalena, 2015). To investigate if short telomeres trigger type I inflammation and accelerated aging through the cGAS-STING pathway, we combined *sting* zebrafish loss of function mutants (*sting*^sa35634/sa35634^, hereby referred to as *sting-/-*) with telomerase-deficient zebrafish (*tert*^hu3430/hu3430^ or *tert-/-*) extensively characterized in our previous studies (El Maï et al, 2020, 2023; Lex et al, 2020; Henriques et al, 2013; Carneiro et al, 2016). Upon incross of double heterozygous fish, we investigated if first generation (G1) *tert-/- sting-/-* double mutants had short telomeres comparable to their *tert-/-* siblings. We observed that, by 9 months of age, *tert-/- sting-/-* had similar mean telomere length to *tert-/-* single mutants in the skin and intestine (Fig. 1A–D; Appendix Fig. S1A–D). In testis, *tert-/- sting-/-* zebrafish had slightly shorter telomeres than *tert-/-* (Fig. 1B; Appendix Fig. S1B) and slightly longer telomeres in the kidney marrow (Fig. 1C; Appendix Fig. S1C). Overall, not only telomere length was similar between *tert-/-* and *tert-/- sting-/-* but were significantly shorter than WT and *sting-/-* siblings.

Previous studies using cell lines reported that short and dysfunctional telomeres lead to the formation of MN (Glück et al, 2017; Dou et al, 2017; Yang et al, 2017). To test whether we would observe an increase in MN upon telomere shortening in zebrafish, we derived fibroblasts from the skin of 9-month-old animals. We observed a tenfold increase in MN formation in *tert-/-* zebrafish when compared to the WT and *sting-/-* siblings. Strikingly, *tert-/- sting-/-* had similar levels of MN as *tert-/-* siblings (Fig. 1E). Thus, telomere shortening results in MN accumulation in vivo.

Recent data from telomerase knockout (Terc−/−) mice show that critically short telomeres modulated the expression of transposons (TE) at subtelomeric regions, by promoting changes in chromatin accessibility and H3K9me3 profiling (Zhao et al, 2023). In our RNAseq analyses, we observe a general mobilization of TE spanning different proliferative tissues (Appendix Fig. S3A,B). Multidimensional scaling (MDS) plots of TE expression were non-overlapping for WT and *tert−/−* mutant in all tissues analyzed (Appendix Fig. S3A). Telomerase-deficient fish showed a clear upregulation of LTR and DNA transposons in testis and kidney marrow and, to a lesser extent, in the gut. We therefore tested specifically the methylation status of Histone H3 (Fig. 1F). In the skin, H3K9me3 is significantly downregulated and, associated to this downregulation, we observe an increase in LTR2 element, confirming transposon derepression (Fig. 1F,G).

After confirming the presence of shorter telomeres, MN and TE mobilization in *tert-/-* and *tert-/- sting-/-* zebrafish, we investigated if the cGAS-STING pathway was active in vivo by quantifying its downstream targets. Therefore, using skin of 9-month-old zebrafish, we analyzed the phosphorylation status of zebrafish Tbk1 and Irf3 and the transcription level of two members of type I interferon response, *isg15* and *ifn-i* (Fig. 2A–C). Comparing *tert-/-* to WT zebrafish, we observed a ca. threefold increase in p-Irf3 and twofold increase in p-Tbk1 (Fig. 2A). Consistently, the expression of *isg15* and *ifn-i* was increased by 10- and 2.5-fold, respectively (Fig. 2B,C). However, the phosphorylation profile of Tbk1 and Irf3 in *tert-/- sting-/-* mutants was similar to their WT and *sting-/-* siblings (Fig. 2A). *tert-/- sting-/-* mutants also lacked type I interferon response, as observed by the reduced levels of *isg15* and *ifn-i* (Fig. 2B,C).

We expanded our study to other proliferative tissues like testis, kidney marrow, and intestine since they all show signs of inflammation in aging zebrafish (Carneiro et al, 2016; El Maï et al, 2023, 2020). Comparing transcription levels of *tert-/-* fish to WT fish, we observed a 15-fold increase in expression for *isg15* and *ifn-i* in testis; a 20-fold increase in *isg15* and a fivefold increase in kidney marrow; and, lastly, a 7.5-fold increase in *isg15* transcript and 2.5-fold increase in *ifn-i* in the intestine. Importantly, inactive cGAS-STING pathway rescued the *isg15* and *ifn-i* levels of *tert-/- sting-/-* to those observed in WT zebrafish in all tissues analyzed (Fig. 2B,C).

To further confirm our data, we used second-generation (G2) *tert-/-* fish. These fish show extremely short telomeres and recapitulate several phenotypes of older G1 fish in only two weeks of life (Anchelin et al, 2013; Henriques et al, 2013; El Maï et al, 2020; Lex et al, 2020). Therefore, we tested whether the interferon response was also altered in these fish. We observed a significant transcriptional activation of *ifn-i* and *isg-15* observed in G2 *tert-/-* fish compared to WT fish (Appendix Fig. S2B). Interestingly, similarly to G1 fish, *sting* mutation did not alter telomere shortening, but rescued the transcriptional activation of interferon response (Appendix Fig. S2A,B). These results show that, despite the presence of shorter telomeres and MN, the cGAS-STING pathway is inactive in *tert-/- sting-/-* mutants.

TE transcription leads to robust activation of RIG-I and MDA5 that recognize dsRNA and ssRNA and initiate a signaling cascade, involving MAVS oligomerization, IRF3/IRF7 activation and the expression of type I interferon response. We therefore investigated whether the expressions of the RNA sensors *rig-I*, *mda5* and *mavs* were increased in telomerase-deficient animals. Similar to TE expression, *rig-I* was overexpressed in *tert-/-* kidney marrow, compared to WT animals, but not in the intestine (Appendix Fig. S2C,D). Recently, the Karlseder lab showed that both DNA sensing (cGAS-STING) and RNA sensing (ZBP1-MAVS) innate

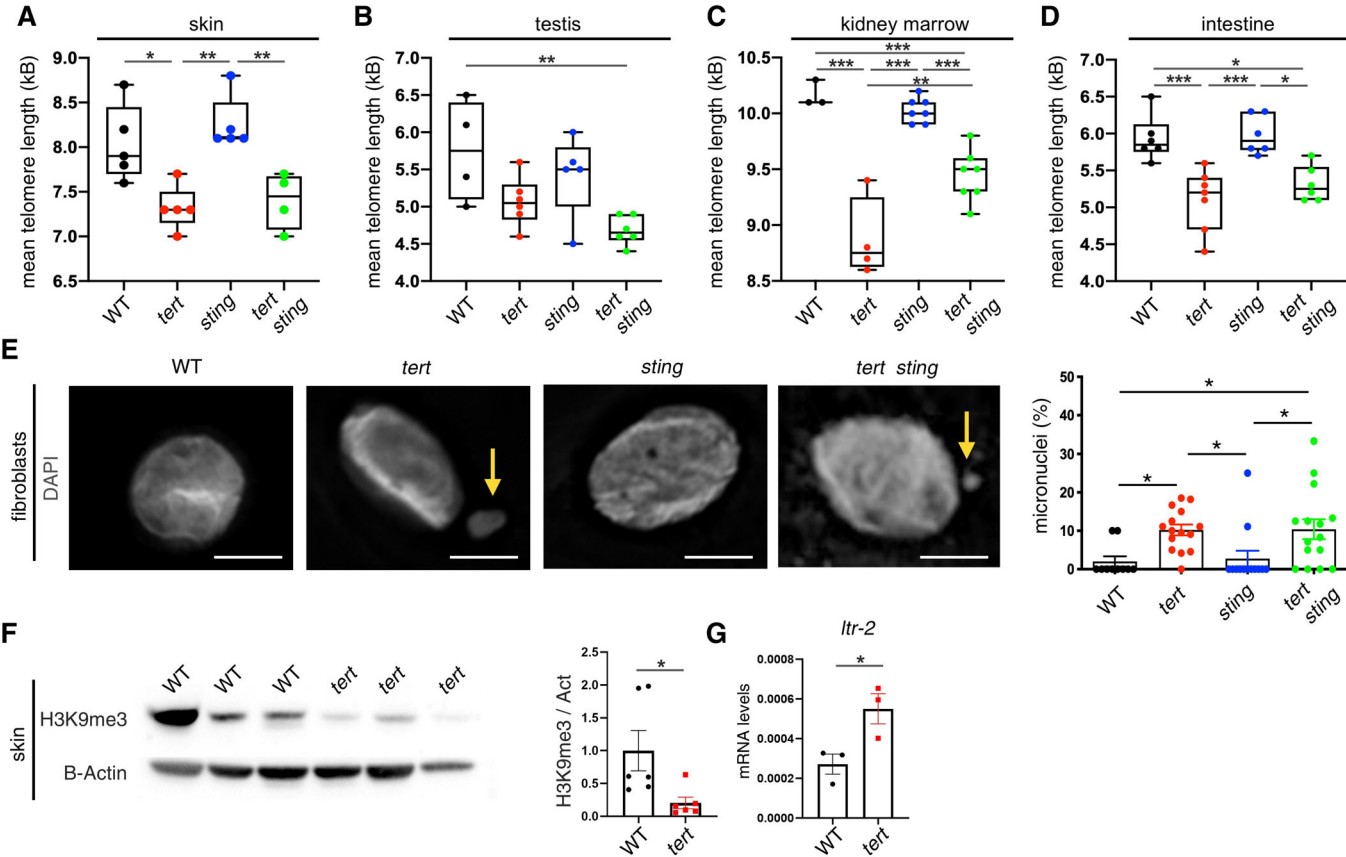

**Figure 1. Telomere shortening induces micronuclei formation and transposon mobilization.**

(A) Quantification of mean telomere length measured by TRF analysis in the skin ($n_{WT} = 5$, $n_{tert-/-} = 5$, $n_{sting-/-} = 5$, $n_{tert-/- sting-/-} = 4$, WT vs tert-/- p = 0.017, tert-/- vs sting-/- p = 0.002, sting-/- vs tert-/- sting-/- p = 0.008). (B) Quantification of mean telomere length measured by TRF analysis in the testis ($n_{WT} = 4$, $n_{tert-/-} = 6$, $n_{sting-/-} = 5$, $n_{tert-/- sting-/-} = 6$, WT vs tert-/- sting-/- p = 0.008). (C) Quantification of mean telomere length measured by TRF analysis in the kidney marrow ($n_{WT} = 3$, $n_{tert-/-} = 4$, $n_{sting-/-} = 7$, $n_{tert-/- sting-/-} = 7$, WT vs tert-/- p = 0.000003, WT vs tert-/- sting-/- p = 0.0008, tert-/- vs sting-/- p = 0.000001, sting-/- vs tert-/- sting-/- p = 0.0005, tert-/- vs tert-/- sting-/- p = 0.003). (D) Quantification of mean telomere length measured by TRF analysis in the intestine ($n_{WT} = 6$, $n_{tert-/-} = 6$, $n_{sting-/-} = 6$, $n_{tert-/- sting-/-} = 6$, WT vs tert-/- p = 0.0007, WT vs tert-/- sting-/- p = 0.015, tert-/- vs sting-/- p = 0.0004, sting-/- vs tert-/- sting-/- p = 0.008). In each boxplot (A–D), the boxes represent the interquartile range (IQR; 25th to 75th percentile). Inside each box, the horizontal line indicates the median. The whiskers extend to the most extreme data points (minima and maxima) within 1.5 × IQR from the quartiles. (E) Representative immunofluorescence images and quantifications of MN formation in the fibroblasts derived from skin ($n_{WT} = 1$, $n_{tert-/-} = 1$, $n_{sting-/-} = 1$, $n_{tert-/- sting-/-} = 1$, WT vs tert-/- p = 0.040, WT vs tert-/- sting-/- p = 0.034, tert-/- vs sting-/- p = 0.047, sting-/- vs tert-/- sting-/- p = 0.040). The yellow arrows point at the micronuclei. Scale bar = 5 μm. (F) Representative western blot images and quantification of H3K9me3 in the skin ($n_{WT} = 6$, $n_{tert-/-} = 6$, WT vs tert-/- p = 0.032). (G) RT-qPCR analysis of ltr-2 gene expression ($n_{WT} = 3$, $n_{tert-/-} = 3$, WT vs tert-/- p = 0.038). Data in Fig. 1E–G are presented as the mean ± s.e.m.; *p < 0.05; **p < 0.01, ***p < 0.001, using a one-way ANOVA and post hoc Tukey test (panel A–E), or unpaired t-test (panel F, G). Source data are available online for this figure.

immunity pathways orchestrate telomere replicative (M2) crisis (Nassour et al, 2023). Their model proposes that short telomeres and MN trigger cGAS-STING, whereas ZBP1 senses the telomeric lncRNA TERRA and activates MAVS. Since both pathways result in the activation of type I interferon response, we tested whether ZBP1-MAVS could also contribute to IFN expression. As zebrafish lack a clear *zbp1* gene orthologue, we relied solely on expression of *mavs* for this analysis. *mavs* mRNA was significantly increased in the testis, but not in the kidney marrow, and intestine (Appendix Fig. S2D). Thus, we observed that telomere shortening triggers multiple responses that include TE expression deregulation and inconsistent activation of the RNA sensor pathway. Therefore, in contrast to *sting*, our results suggest that *mavs* expression is not required for interferon response upon telomere shortening.

## cGAS-STING is required for increased p53 levels in the presence of DNA damage

We previously reported that telomere shortening leads to increased p53 levels as a consequence of activation of DDR in zebrafish (El Maï et al, 2020, 2023; Henriques et al, 2013; Carneiro et al, 2016). To identify the cause for activation of cGAS-STING in prematurely aged *tert-/-* mutants, we analyzed DNA damage and activation of DDR in 9-month-old animals. First, we quantified the number of phosphorylated H2AX (γ-H2AX) stained cells by immunofluorescence in the tissues of interest. As in our previous work (El Maï et al, 2023), γ-H2AX staining was dispersed through the nucleoplasm and aggregated in foci (Fig. 3A insets). *tert-/-* zebrafish displayed a threefold increase of γ-H2AX stained cells in the skin (Fig. 3B), fourfold in the testis (Fig. 3C), and a fivefold increase in both the kidney, marrow (Fig. 3D), and the intestine (Fig. 3E)

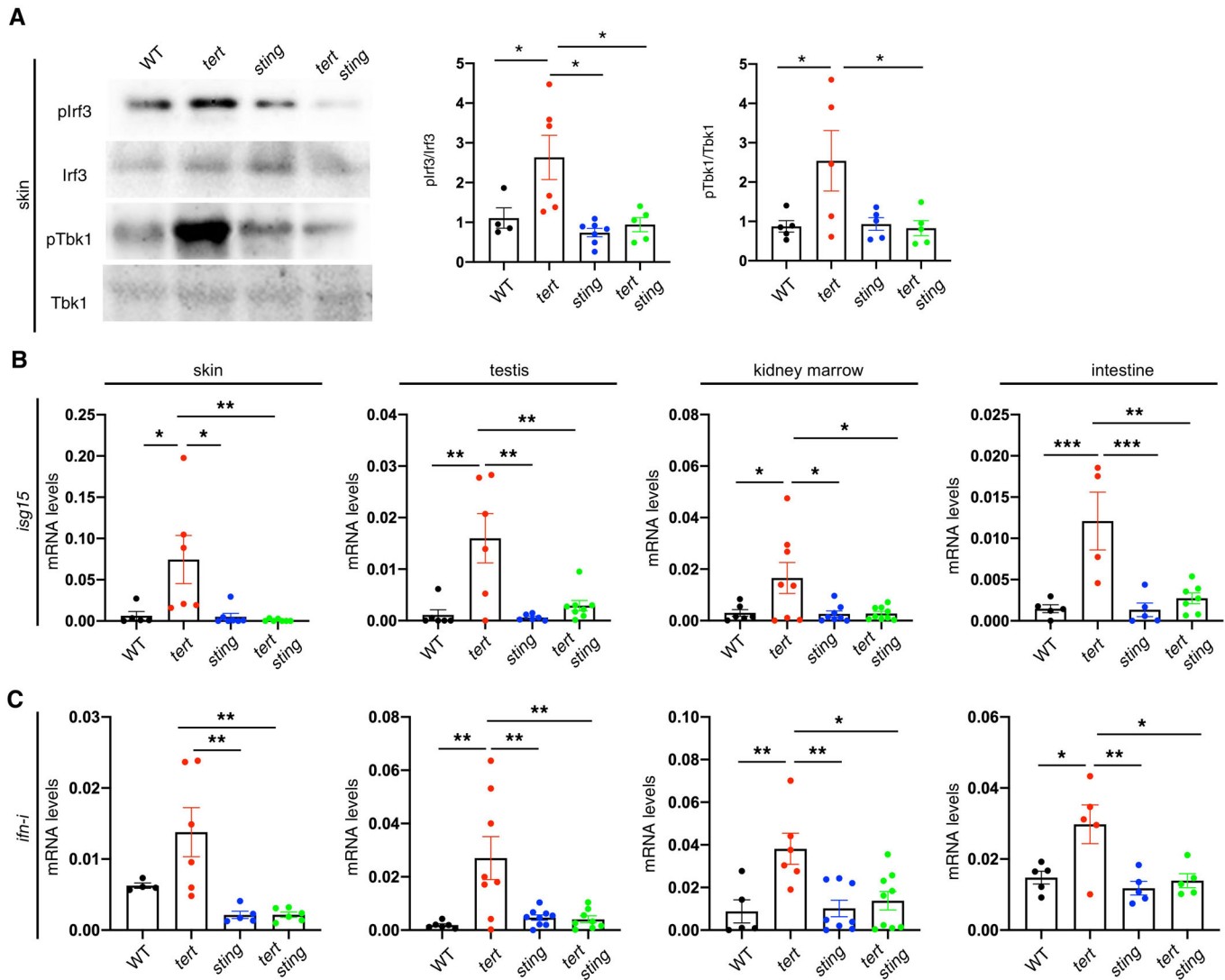

**Figure 2. Telomere shortening activates the cGAS-STING pathway.**

(A) Representative western blot images and quantification of downstream targets of cGAS-STING pathway ($n_{WT}$ = 4–5, $n_{tert-/-}$ = 5–6, $n_{sting-/-}$ = 5–7, $n_{tert-/- sting-/-}$ = 5, p-Irf3/Irf3: WT vs tert-/- p = 0.049, tert-/- vs tert-/- sting-/- p = 0.011; p-Tbk1/Tbk1: WT vs tert-/- p = 0.035, tert-/- vs sting-/- p = 0.002, tert-/- vs tert-/- sting-/- p = 0.042). (B) RT-qPCR analysis of isg15 gene expression in the skin, testis, kidney marrow and intestine ($n_{WT}$ = 5–6, $n_{tert-/-}$ = 4–8, $n_{sting-/-}$ = 5–8, $n_{tert-/- sting-/-}$ = 7–9, skin: WT vs tert-/- p = 0.024, sting-/- vs tert-/- p = 0.011, N = 5–7, tert-/- vs tert-/- sting-/- p = 0.007; testis: WT vs tert-/- p = 0.001, sting-/- vs tert-/- p = 0.001, tert-/- vs tert-/- sting-/- p = 0.002; kidney marrow: N = 6–9, WT vs tert-/- p = 0.045, sting-/- vs tert-/- p = 0.022, tert-/- vs tert-/- sting-/- p = 0.019; intestine: WT vs tert-/- p = 0.0008, sting-/- vs tert-/- p = 0.0007, tert-/- vs tert-/- sting-/- p = 0.001). (C) RT-qPCR analysis of ifn-i gene expression in the skin, testis, kidney marrow and intestine ($n_{WT}$ = 4–6, $n_{tert-/-}$ = 5–8, $n_{sting-/-}$ = 5–8, $n_{tert-/- sting-/-}$ = 6–9, skin: sting-/- vs tert-/- p = 0.004, tert-/- vs tert-/- sting-/- p = 0.002; testis: WT vs tert-/- p = 0.003, sting-/- vs tert-/- p = 0.003, tert-/- vs tert-/- sting-/- p = 0.003; kidney marrow: WT vs tert-/- p = 0.008, sting-/- vs tert-/- p = 0.004, tert-/- vs tert-/- sting-/- p = 0.011; intestine: WT vs tert-/- p = 0.020, sting-/- vs tert-/- p = 0.005, tert-/- vs tert-/- sting-/- p = 0.014). Data were presented as the mean ± s.e.m.; *p < 0.05; **p < 0.01, ***p < 0.001, using a one-way ANOVA and post hoc Tukey test. Source data are available online for this figure.

when compared to WT and sting-/- mutants. Contrarily, tert-/- sting-/- mutants showed similar levels of γ-H2AX compared to the tert-/- siblings (Fig. 3A–D). Thus, consistent with shorter telomere length in tert-/- and tert-/- sting-/- mutants, we observed similar levels of the DNA damage marker γ-H2AX in these tissues.

Increased γ-H2AX levels in tert-/- zebrafish is accompanied by elevation in p53 protein levels (El Maï et al, 2023, 2020). tert-/- zebrafish exhibited an increase of fivefold (Fig. 3F,G) in the skin and testis (Fig. 3F–H), threefold in the kidney marrow (Fig. 3F,I), and sixfold in the intestine (Fig. 3F,J) when compared to WT and

sting-/- mutants (Fig. 3E–H). Surprisingly, p53 levels in the tert-/- sting-/- mutants were similar to the WT and sting-/- siblings (Fig. 3F–J). We also investigated the involvement of the RNA sensor mavs in the stabilization of p53 using morpholino (MO) gene downregulation experiments in G2 tert-/- zebrafish. Even though we significantly reduced mavs expression using mavs MOs (Appendix Fig. S3F), inhibition of mavs did not result in a lower level of p53 in G2 tert-/- fish (Appendix Fig. S3H). Our results indicate that, in contrast to mavs expression, a functional cGAS-STING is required for elevated p53 levels in response to telomere

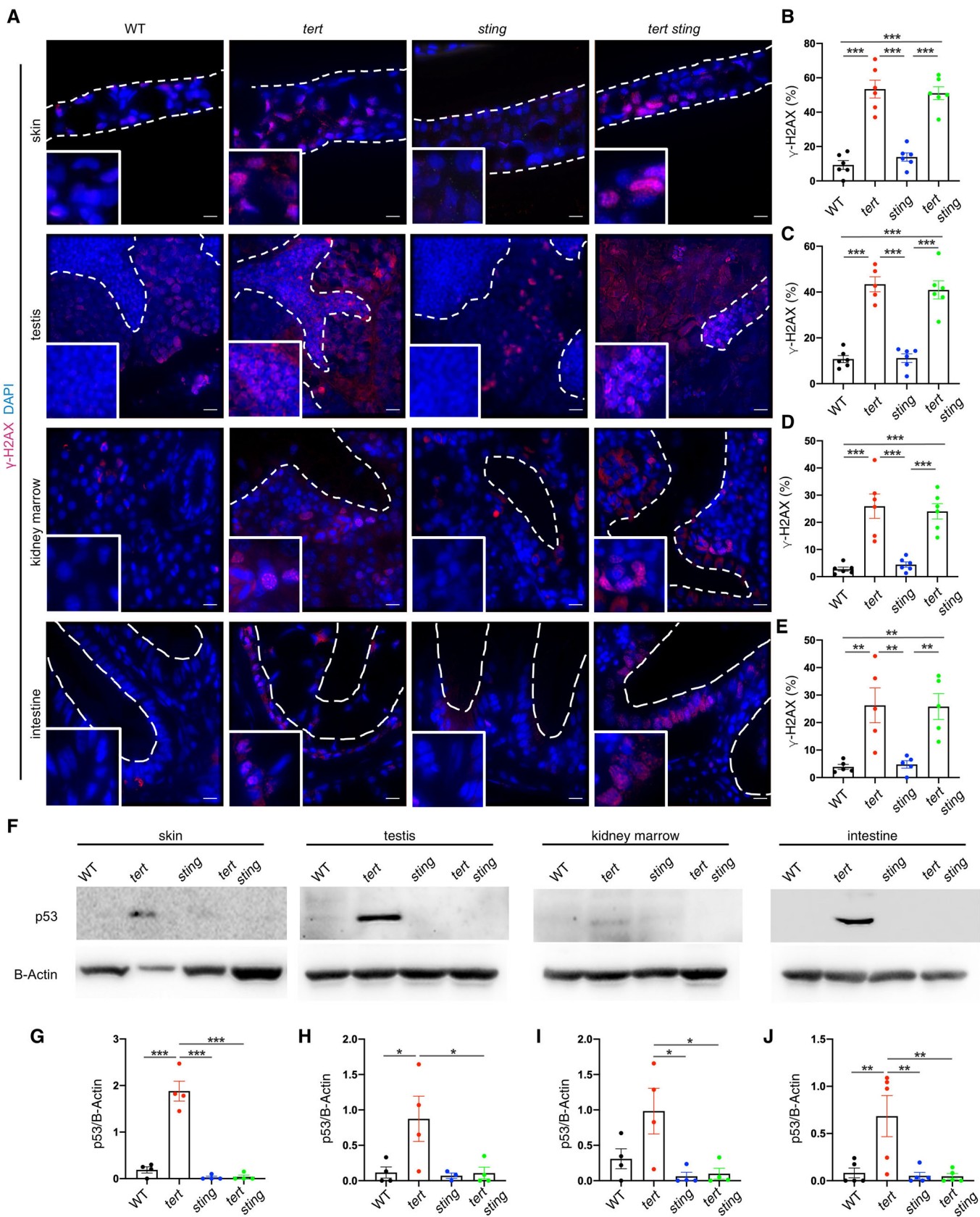

◀ **Figure 3. cGAS-STING pathway inactivation attenuates DNA damage response.**

(A) Representative immunofluorescence images of DNA damage. Scale bar = 10 μm. (B) Quantification of DNA damage in skin ($n_{WT}$ = 6, $n_{tert-/-}$ = 6, $n_{sting-/-}$ = 6, $n_{tert-/- \, sting-/-}$ = 6; WT vs tert-/- $p$ = 0.00001, sting-/- vs tert-/- $p$ = 0.000001, *WT vs tert-/- sting-/- $p$ = 0.000001, sting-/- vs tert-/- sting-/- $p$ = 0.000003*). (C) Quantification of DNA damage in testis ($n_{WT}$ = 6, $n_{tert-/-}$ = 6, $n_{sting-/-}$ = 6, $n_{tert-/- \, sting-/-}$ = 6, $N$ = 5–6, WT vs tert-/- $p$ = 0.000001, *sting-/- vs tert-/- $p$ = 0.000001, WT vs tert-/- sting-/- $p$ = 0.000001, sting-/- vs tert-/- sting-/- $p$ = 0.000002*). (D) Quantification of DNA damage in kidney marrow ($n_{WT}$ = 6, $n_{tert-/-}$ = 6, $n_{sting-/-}$ = 6, $n_{tert-/- \, sting-/-}$ = 6; WT vs tert-/- $p$ = 0.00004, sting-/- vs tert-/- $p$ = 0.0001, WT vs tert-/- sting-/- $p$ = 0.0001, sting-/- vs tert-/- sting-/- $p$ = 0.0003). (E) Quantification of DNA damage in intestine ($n_{WT}$ = 5, $n_{tert-/-}$ = 5, $n_{sting-/-}$ = 5, $_{tert-/- \, sting-/-}$ = 5; WT vs tert-/- $p$ = 0.006, sting-/- vs tert-/- $p$ = 0.008, WT vs tert-/- sting-/- $p$ = 0.007, sting-/- vs tert-/- sting-/- $p$ = 0.0095). (F) Representative western blot images of p53. (G) Quantification of p53 levels in the skin ($n_{WT}$ = 4, $n_{tert-/-}$ = 4, $n_{sting-/-}$ = 4, $n_{tert-/- \, sting-/-}$ = 4, WT vs tert-/- $p$ = 0.000001, sting-/- vs tert-/- $p$ = 0.000001, tert-/- vs tert-/- sting-/- $p$ = 0.000001). (H) Quantification of p53 levels in the testis ($n_{WT}$ = 4, $n_{tert-/-}$ = 4, $n_{sting-/-}$ = 4, $n_{tert-/- \, sting-/-}$ = 4, WT vs tert-/- $p$ = 0.050, tert-/- vs tert-/- sting-/- $p$ = 0.048). (I) Quantification of p53 levels in the kidney marrow ($n_{WT}$ = 4, $n_{tert-/-}$ = 4, $n_{sting-/-}$ = 4, $n_{tert-/- \, sting-/-}$ = 4, sting-/- vs tert-/- $p$ = 0.017, tert-/- vs tert-/- sting-/- $p$ = 0.022). (J) Quantification of p53 levels in the intestine ($n_{WT}$ = 5, $n_{tert-/-}$ = 5, $n_{sting-/-}$ = 5, $n_{tert-/- \, sting-/-}$ = 5, WT vs tert-/- $p$ = 0.009, sting-/- vs tert-/- $p$ = 0.006, tert-/- vs tert-/- sting-/- $p$ = 0.006). Data were presented as the mean ± s.e.m.; *$p$ < 0.05; **$p$ < 0.01, ***$p$ < 0.001, using a one-way ANOVA and post hoc Tukey test. Source data are available online for this figure.

shortening. This is in agreement with studies showing that p53 expression and stability are regulated by IFNs (Huang et al, 2014; Takaoka et al, 2003).

## cGAS-STING is required for senescence and SASP caused by telomere shortening

Given that p53 was not elevated in aging *tert-/- sting-/-* zebrafish, despite the shorter telomere length, we decided to assess the remaining phenotypes linked with *tert-/-* premature aging. Previous studies in human cells showed that cGAS-STING is required for cell senescence (Glück et al, 2017; Dou et al, 2017; Yang et al, 2017). Using 9-month-old zebrafish, we studied cell senescence using the SA-Beta-Galactosidase (SA-B-gal) assay in addition to expression of *cdkn2a/b* (p15/16) and *cdkn1a* (p21) by RT-qPCR. As previously observed (Henriques et al, 2013; El Maï et al, 2020, 2023; Carneiro et al, 2016), proliferative tissues of *tert-/-* mutants were strongly stained with SA-B-gal, but not those of WT or *sting-/-* siblings (Fig. 4A). Consistently, we confirmed an elevated expression of *cdkn2a/b* and *cdkn1a* senescence markers in *tert-/-* mutants (Fig. 4B–D). Senescence in *tert-/-* was accompanied by expression of SASP-related genes, namely *il1b*, *tgfb1b* and *mmp15a*. As expected, none of these genes were elevated in WT or *sting-/-* siblings (Fig. 4B–D).

With lower levels of p53, cell senescence was also reduced in *tert-/- sting-/-* tissues, as observed by low SA-Beta-gal and expression of *cdkn2a/b* and *cdkn1a*. In the absence of senescence, expression of SASP factors were also reduced in *tert-/- sting-/-* mutants to the ones observed in WT (Fig. 4B–E).

Similarly, G2 *tert-/-* fish showed an increase in senescence markers and SASP. However, *sting* mutation (but not *mavs* downregulation), was able to reverse these phenotypes in G2 *tert-/-* larvae (Appendix Figs. S2C and S3G). Thus, in agreement with previous in vitro studies, our results indicate that cGAS-STING is required for senescence and SASP of proliferative tissues in vivo.

## cGAS-STING controls cell proliferation and tissue integrity of *tert-/-* zebrafish

Replicative cell senescence is a barrier against cell proliferation in response to telomere shortening. We thus investigated whether the absence of senescence in *tert-/- sting-/-* mutants would result in increased cell proliferation. We examined proliferative tissues by immunofluorescence using antibodies against PCNA, a marker for cell proliferation. As previously reported (Henriques et al, 2013; Carneiro et al, 2016; El Maï et al, 2023), cell proliferation of *tert-/-*

zebrafish is significantly reduced in proliferative tissues when compared to WT and *sting-/-* siblings. However, lower cell proliferation was recovered to WT levels in *tert-/- sting-/-* zebrafish (Fig. 5A and quantification in Fig. 5B–E). Thus, with lower cell senescence and p53 levels, the absence of cGAS-STING results in an increase in cell proliferation despite telomere shortening in telomerase-deficient zebrafish.

Previously, we showed that telomere shortening impacts tissue integrity and results in morphological defects of several tissues (El Maï et al, 2023, 2020; Henriques et al, 2013; Carneiro et al, 2016). Inflammation of the intestine causes an increase in the thickness of the *lamina propria* in *tert-/-* zebrafish (mean of 17.5 μm, Fig. 5F,G). However, the width of the *lamina propria* in *tert-/- sting-/-* zebrafish was similar to the WT and *sting-/-* siblings (Fig. 5G). Loss of gut tissue integrity activates the YAP-TAZ pathway in aging *tert-/-* zebrafish (El Maï et al, 2023). In agreement, we observed an increase of *ctfg* and *cyr61* levels, targets of YAP-TAZ pathway (Fig. 5H), in *tert-/-* zebrafish compared to WT and *sting-/-* siblings. However, expressions of YAP-TAZ target genes were reduced to the WT levels in the *tert-/- sting-/-* zebrafish (Fig. 5H). Similarly, the remaining proliferative tissues also showed increased expression of *ctgf* and *cyr61* in *tert-/-* zebrafish that were absent in *tert-/- sting-/-* siblings (testis: 2-fold, Fig. 5I, skin: 2.5-fold and 5-fold, respectively, kidney marrow: ~10-fold, Appendix Fig. S5B–D). Thus, our results suggest that tissue integrity of the intestine and other tissues caused by telomere shortening is rescued by an increase in cell proliferation upon inactivation of the cGAS-STING pathway.

## cGAS-STING causes premature aging of *tert-/-* zebrafish

Given the previous results at the cellular level, we investigated the consequences to the whole organism of the absence of cGAS-STING and Type I interferon in response to telomere shortening. We previously established male fertility and testicular atrophy as robust assays for aging zebrafish (Henriques et al, 2013; Carneiro et al, 2016; Şerifoğlu et al, 2024). To measure testicular atrophy, we quantified the mature sperm area in HE stained sections of whole testis (Fig. 5J). The percentage of mature sperm area of 9-month-old *tert-/-* zebrafish was reduced to ~5%, compared to ~50% in WT and *sting-/-* siblings (Fig. 5K). However, the percentage of mature sperm area increased to ~40% in *tert-/- sting-/-* zebrafish (Fig. 5K). Denoting the observed loss of tissue integrity in testis in *tert-/-* prematurely aged zebrafish, we saw increased expression of the YAP-TAZ pathway targets in *ctgf* and *cyr61* compared to WT and *sting-/-* siblings (Fig. 5I). Consistent with our previous results, expression of *ctfg* and *cyr61* were reduced in *tert-/- sting-/-* to WT levels (Fig. 5I).

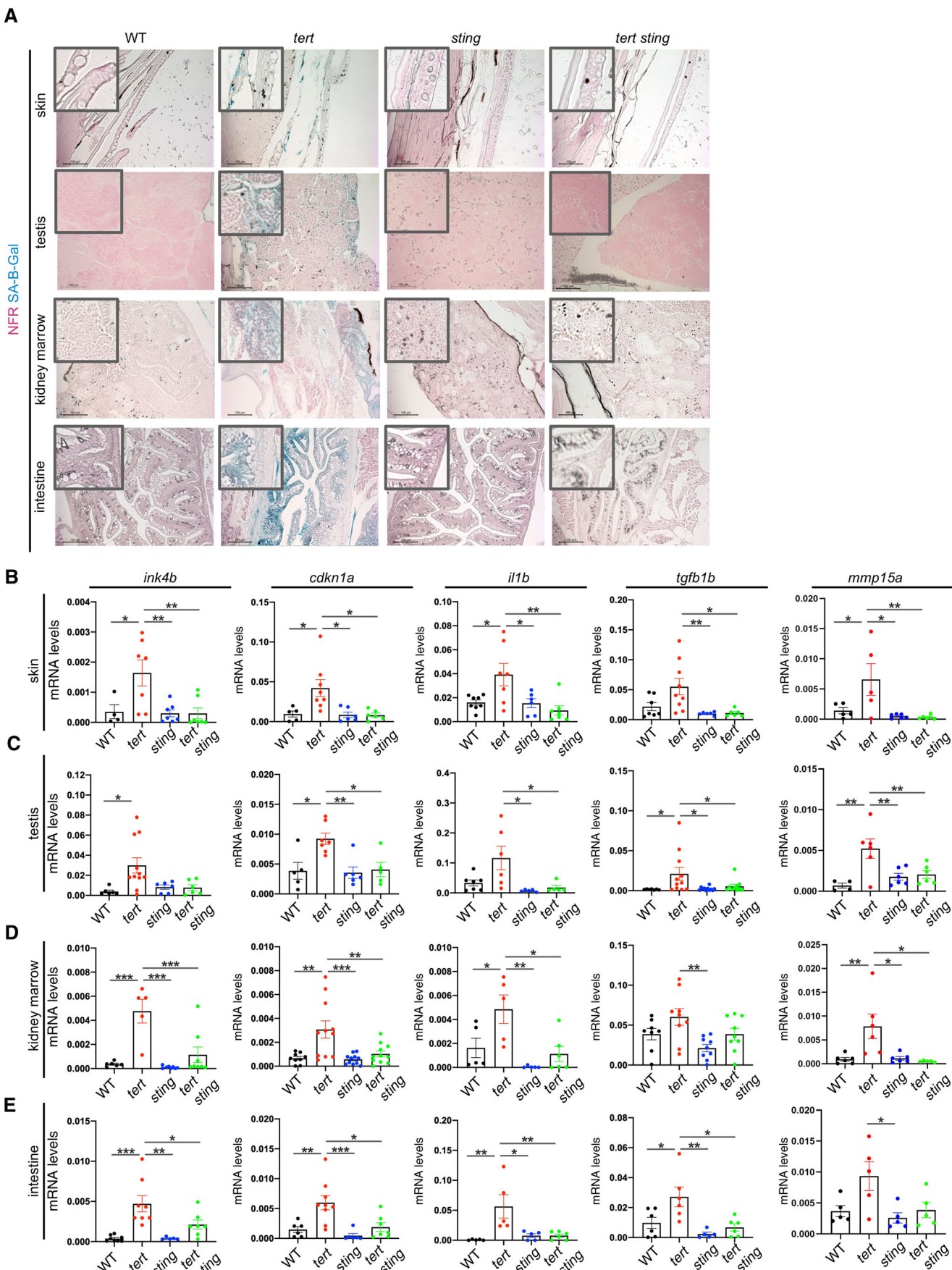

**Figure 4.   SASP induced by short telomeres are controlled by the cGAS-STING pathway.**

(A) Representative images of senescence-associated beta galactosidase staining in skin, testis, kidney, marrow and intestine ($n_{WT}$ = 3, $n_{tert-/-}$ = 3, $n_{sting-/-}$ = 3, $n_{tert-/-\ sting-/-}$ = 3). Scale bar = 100 μm. (B) RT-qPCR analysis of inflammatory markers and SASP factors in the skin ($n_{WT}$ = 4–8, $n_{tert-/-}$ = 5–8, $n_{sting-/-}$ = 6–7, $n_{tert-/-\ sting-/-}$ = 5–7; cdkn2a/b: WT vs tert-/- p = 0.035, sting-/- vs tert-/- p = 0.009, tert-/- vs tert-/- sting -/- p = 0.004; cdnk1a: WT vs tert-/- p = 0.025, sting-/- vs tert-/- p = 0.014, tert-/- vs tert-/- sting-/- p = 0.013; il1b: WT vs tert-/- p = 0.023, sting-/- vs tert-/- p = 0.031, tert-/- vs tert-/- sting-/- p = 0.004,; tgf1b: sting-/- vs tert-/- p = 0.014, tert-/- vs tert-/- sting-/- p = 0.017; mmp15a: sting-/- vs tert-/- p = 0.011, tert-/- vs tert-/- sting-/- p = 0.009). (C) RT-qPCR analysis of inflammatory markers and SASP factors in the testis ($n_{WT}$ = 4-7, $n_{tert-/-}$ = 6–11, $n_{sting-/-}$ = 5–7, $n_{tert-/-\ sting-/-}$ = 5–8; cdkn2a/b: WT vs tert-/- p = 0.0234; cdnk1a: WT vs tert-/- p = 0.013, tert-/- vs tert-/- sting-/- p = 0.018; il1b: tert-/- vs tert-/- sting-/- p = 0.030; tgf1b: WT vs tert-/- p = 0.029, sting-/- vs tert-/- p = 0.020, tert-/- vs tert-/- sting-/- p = 0.043; mmp15a: WT vs tert-/- p = 0.004, sting-/- vs tert-/- p = 0.016, tert-/- vs tert-/- sting-/- p = 0.028). (D) RT-qPCR analysis of inflammatory markers and SASP factors in the kidney marrow ($n_{WT}$ = 5–9, $n_{tert-/-}$ = 5–11, $n_{sting-/-}$ = 6–11, $n_{tert-/-\ sting-/-}$ = 6–10; cdkn2a/b: WT vs tert-/- p = 0.0001, sting-/- vs tert-/- p = 0.00003, tert-/- vs tert-/- sting-/- p = 0.0006; cdnk1a: WT vs tert-/- p = 0.002, sting-/- vs tert-/- p = 0.0005, tert-/- vs tert-/- sting-/- p = 0.005; il1b: WT vs tert-/- p = 0.047, sting-/- vs tert-/- p = 0.003, tert-/- vs tert-/- sting-/- p = 0.014; tgf1b: sting-/- vs tert-/- p = 0.005; mmp15a: WT vs tert-/- p = 0.009, sting-/- vs tert-/- p = 0.011, tert-/- vs tert-/- sting-/- p = 0.008). (E) RT-qPCR analysis of inflammatory markers and SASP factors in the intestine ($n_{WT}$ = 5–7, $n_{tert-/-}$ = 5–8, $n_{sting-/-}$ = 5–6, $n_{tert-/-\ sting-/-}$ = 5–7; cdkn2a/b: WT vs tert-/- p = 0.0004, sting-/- vs tert-/- p = 0.001, tert-/- vs tert-/- sting-/- p = 0.038; cdnk1a: WT vs tert-/- p = 0.007, sting-/- vs tert-/- p = 0.0009, tert-/- vs tert-/- sting-/- p = 0.011; il1b: WT vs tert-/- p = 0.004, sting-/- vs tert-/- p = 0.012, tert-/- vs tert-/- sting-/- p = 0.009; tgf1b: WT vs tert-/- p = 0.034, sting-/- vs tert-/- p = 0.003; mmp15a: WT vs tert-/- NS p = 0.060, sting-/- vs tert-/- p = 0.021, tert-/- vs tert-/- sting-/- p = 0.070). All data are presented as the mean ± s.e.m.; *p < 0.05; **p < 0.01, ***p < 0.001, using a one-way ANOVA and post hoc Tukey test. Source data are available online for this figure.

We next asked if the rescue in the testis morphology of tert-/- sting-/- would restore fertility of aging tert-/- zebrafish. We previously reported that, from 6-month-old, tert-/- males become infertile, paralleling the loss of fertility observed in 18-month-old WT fish (Carneiro et al, 2016). As expected, 9-month-old tert-/- males were unable to produce fertilized eggs when crossed with young WT females (Fig. 6A). Strikingly, tert-/- sting-/- males were still fertile, even if slightly lower than the WT and sting-/- siblings (Fig. 6A), losing completely their fertility at 13 months old (Appendix Fig. S5). Our data shows that inhibition of cGAS-STING and type I interferon response is sufficient to prolong fertility in aging tert-/- zebrafish.

Like other vertebrates, the incidence of cancer increases with age in zebrafish (Carneiro et al, 2016; Şerifoğlu et al, 2024; Spitsbergen et al, 2012). Similar to other age-associated phenotypes (Fig. 6B), the spontaneous cancer incidence is accelerated in younger ages in tert-/- zebrafish (Şerifoğlu et al, 2024; Carneiro et al, 2016). We quantified the rate of spontaneous tumor formation, mostly seminomas, in male zebrafish. We observed that tert-/- zebrafish developed tumors from the age of 11 month old and by the age of 17 months (Fig. 6C). Strikingly, tert-/- sting-/- mutants developed macroscopic tumors around the age of 12 months but were restricted to 5% of the population (Fig. 6C). Importantly, spontaneous cancer incidence of tert-/- sting-/- zebrafish was not statistically different from the WT and sting-/- siblings.

Other phenotypes of aging, such as kyphosis (abnormal curvature of the spine), caused by increased weakness of the spinal bones, and cachexia (excessive muscle wasting), caused by muscle tissue atrophy, are present in younger tert-/- mutants and only appear later in WT zebrafish (Henriques et al, 2013; Carneiro et al, 2016). By the age of 15 months, 60% of tert-/- zebrafish showed aging phenotypes (Fig. 6B) and weighed significantly less (Fig. 6D) than WT and sting-/- siblings. However, the incidence of aging phenotypes ameliorated in tert-/- sting-/- mutants (Fig. 6B) and weight was restored to WT levels (Fig. 6D).

Finally, we compared the lifespan of tert-/- mutants to their tert-/- sting-/- siblings. Whereas tert-/- mutants had a mean lifespan of 17 months, lifespan was extended to 24 months in tert-/- sting-/- mutants (Fig. 6E). The mean lifespan of tert-/- sting-/- zebrafish was not statistically different from WT and sting-/- siblings (Fig. 6E). Consistently, sting mutation in the G2 tert-/- fish was able to significantly rescue larval lifespan (Appendix Fig. S2E). Overall, our results show that by inhibiting cGAS-STING and, consequently, type I interferon, we observed an increase in lifespan of a prematurely aging

vertebrate model by 41%. More importantly, we recover most age-associated phenotypes of aging tert-/- mutants, increasing their healthspan without the increase in cancer incidence.

# Discussion

Aging is accompanied by a wide range of physiological changes, such as chronic inflammation (López-Otín et al, 2023). Age-associated inflammation in the absence of overt infection has been termed inflammaging (Franceschi et al, 2018). While the origins of inflammaging are mostly unclear, it is typically characterized by high levels of pro-inflammatory cytokines, chemokines, acute phase proteins, and soluble cytokine receptors in the serum (Franceschi et al, 2018; Li et al, 2023). Inflammaging was shown to contribute to the development of age-associated diseases, such as neurodegenerative diseases, cardiovascular diseases and cancer (López-Otín et al, 2023). These are the main causes of morbidity and mortality in the elderly. Recent exciting new data revealed that DNA damage and inflammation are connected by the cGAS-STING pathway (Nassour et al, 2019; Dou et al, 2017; Glück et al, 2017). Moreover, chemical inhibition of STING suppresses aging-associated inflammation and neurodegeneration (Gulen et al, 2023). Our study extends these observations by showing that activation of the cGAS-STING pathway caused by telomere shortening is responsible for premature aging in zebrafish.

What triggers inflammaging upon telomere shortening? We documented several potential triggers for type I interferon responses in our work. We observed an increase in MN in tert-/- zebrafish. However, recent data has shown that, even though cGAS is recruited to MN, the presence of chromatin might not lead to elevated cGAMP and STING activation (Sato and Hayashi, 2024). Further evidence from aging mice showed that mtDNA released from disrupted mitochondria was an important source for cGAS-STING activation (Gulen et al, 2023). In our previous work (El Maï et al, 2020), we showed that telomerase mutants have dysfunctional mitochondria with disrupted membranes, providing a likely source for cytoplasmic mtDNA as an additional trigger. Moreover, mirroring what was previously observed in human fibroblasts (De Cecco et al, 2019; Nassour et al, 2023), we documented a tissue-specific elevation of TE expression, that contribute and likely reinforce the activation type I interferon during aging. Finally, expression of the telomeric lncRNA TERRA could also contribute to the activation of cGAS-STING via the ZBP1/MAVS pathway, as

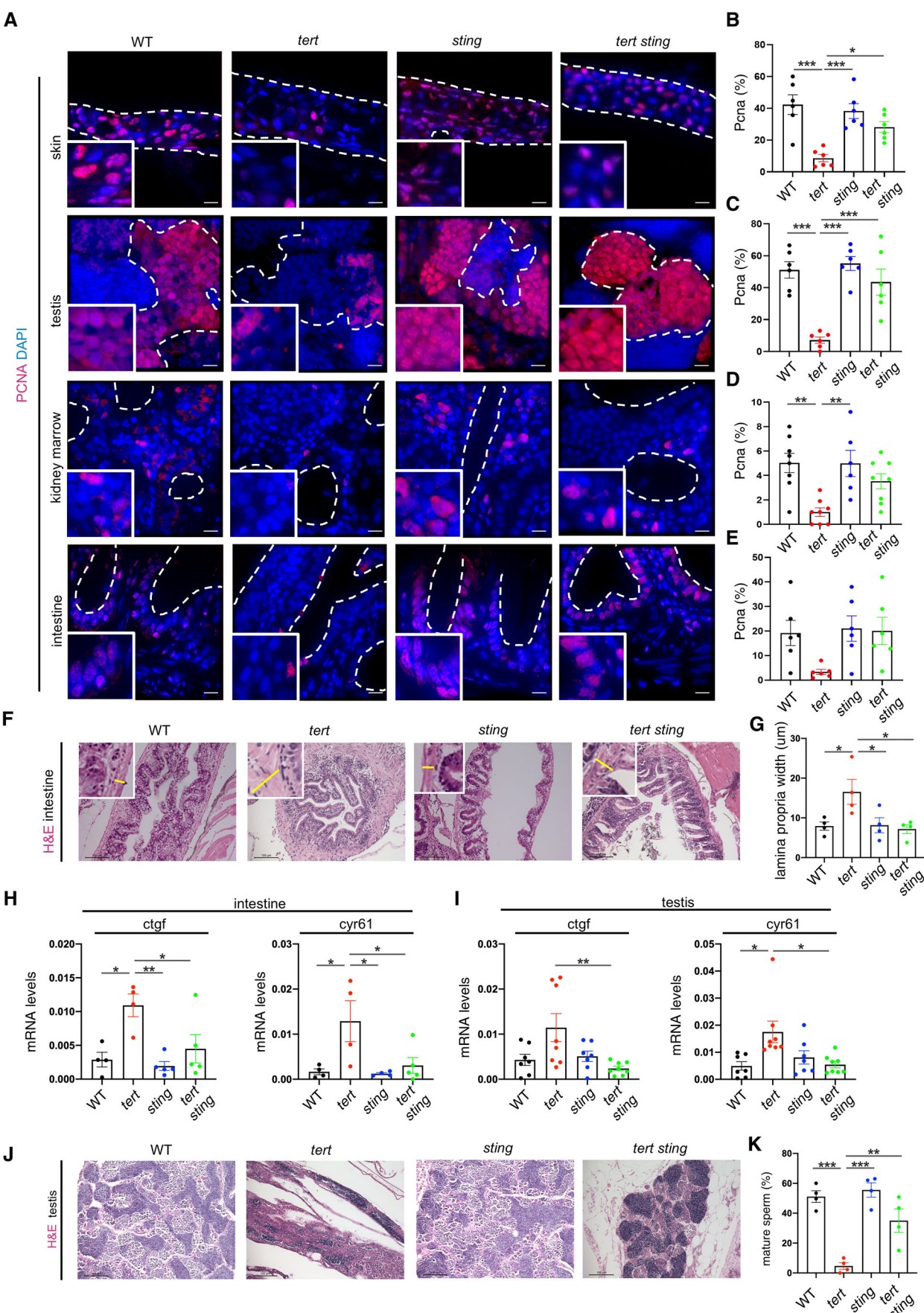

◀ **Figure 5. cGAS-STING pathway control proliferation in telomeric dysfunction.**

(A) Representative immunofluorescence images of proliferation and apoptosis. Scale bar = 10 μm. (B) Quantification of proliferation in the skin ($n_{WT}$ = 5, $n_{tert-/-}$ = 6, $n_{sting-/-}$ = 6, $n_{tert-/- sting-/-}$ = 6 WT vs *tert-/- p* = 0.0001, *sting-/-* vs *tert-/- p* = 0.0006, *tert-/-* vs *tert-/- sting-/- p* = 0.024). (C) Quantification of proliferation in the testis ($n_{WT}$ = 6, $n_{tert-/-}$ = 6, $n_{sting-/-}$ = 6, $n_{tert-/- sting-/-}$ = 6, WT vs *tert-/- p* = 0.00007, *sting-/-* vs *tert-/- p* = 0.00002, *tert-/-* vs *tert-/- sting-/- p* = 0.0006). (D) Quantification of proliferation in the kidney marrow ($n_{WT}$ = 8, $n_{tert-/-}$ = 8, $n_{sting-/-}$ = 6, $n_{tert-/- sting-/-}$ = 8, WT vs *tert-/- p* = 0.002, *sting-/-* vs *tert-/- p* = 0.004). (E) Quantification of proliferation in the intestine ($n_{WT}$ = 6, $n_{tert-/-}$ = 5, $n_{sting-/-}$ = 6, $n_{tert-/- sting-/-}$ = 8). (F) Representative hematoxylin eosin staining of intestine, insets with yellow lines representative of lamina propria thickness. Scale bar = 100 μm. (G) Quantification of lamina propria width ($n_{WT}$ = 4, $n_{tert-/-}$ = 4, $n_{sting-/-}$ = 4, $n_{tert-/- sting-/-}$ = 4, WT vs *tert-/- p* = 0.042, *sting-/-* vs *tert-/- p* = 0.048). (H) RT-qPCR analysis of YAP-TAZ pathway targets in the intestine ($n_{WT}$ = 4, $n_{tert-/-}$ = 4, $n_{sting-/-}$ = 5, $n_{tert-/- sting-/-}$ = 5, ctfg: WT vs *tert-/- p* = 0.016 *sting-/-* vs *tert-/- p* = 0.005, *tert-/-* vs *tert-/- sting-/- p* = 0.044; *cyr61*: WT vs *tert-/- p* = 0.030, *sting-/-* vs *tert-/- p* = 0,023, *tert-/-* vs *tert-/- sting-/- p* = 0.046). (I), RT-qPCR analysis of YAP-TAZ pathway targets in the testis ($n_{WT}$ = 7, $n_{tert-/-}$ = 8, $n_{sting-/-}$ = 6–7, $_{tert-/- sting-/-}$ = 7–8, ctfg: *tert-/-* vs *tert-/- sting-/- p* = 0.008, *cyr61*: *tert-/-* vs *tert-/- sting-/- p* = 0.011). (J) Representative hematoxylin eosin staining of testis. Scale bar = 100 μm. (K) Quantification of mature sperm area ($n_{WT}$ = 4, $n_{tert-/-}$ = 4, $n_{sting-/-}$ = 4, $n_{tert-/- sting-/-}$ = 4, WT vs *tert-/- p* = 0.0002, *sting-/-* vs *tert-/- p* = 0.00007, *tert-/-* vs *tert-/- sting-/- p* = 0.006). All data were presented as the mean ± s.e.m.; *$p < 0.05$; **$p < 0.01$; ***$p < 0.001$, using a one-way ANOVA and post hoc Tukey test. Source data are available online for this figure.

reported for p53-deficient human cells undergoing crisis (Nassour et al, 2023). However, we were unable to detect a clear elevation of *mavs* expression in all tissues. A likely explanation may relate to *tert-/-* zebrafish being p53 proficient and, therefore, their comparatively longer telomeres may not express sufficient levels of TERRA to trigger its response.

Our work reveals that DDR and senescence triggered by short telomeres require an active cGAS-STING pathway. Even though telomere length and γ-H2AX levels were similar between *tert-/-* and *tert-/- sting-/-* zebrafish, p53 was only elevated in *tert-/-* mutants. Although a definite explanation is currently unavailable, we suggest potential leads for this observation. First, IFN-b signaling was shown to induce p53 transcription (Takaoka et al, 2003). Therefore, cGAS-STING may be required to promote p53 expression independently of canonical DDR. Second, the interferon-stimulated gene ISG15, an ubiquitin-like protein, is involved in p53 degradation by the 20S proteasome. ISG15 primarily targets misfolded p53, and deletion of ISG15 results in suppression of p53 activity and functions (Huang et al, 2014). Thus, the absence of cGAS-STING may lead to p53 destabilization. Third, cGAS-STING may sensitize cells to DNA damage by lowering the threshold for DDR activation. In the absence of cGAS-STING, the ATM/ATR-p53 pathway may remain ineffective until genome instability is triggered by telomere-end fusions during crisis. In this scenario, phosphorylation of γ-H2AX could be achieved through parallel pathways, such as DNA-PKcs.

Absence of cGAS-STING in aging telomerase-deficient zebrafish results in low p53 levels, reduced senescence and increased cell proliferation, thus rescuing damage imposed to proliferative tissues. This phenotype is also observed in *tp53-/- tert-/-* double-mutant zebrafish (Şerifoğlu et al, 2024). Similar to late-generation telomerase-deficient mice (Rudolph et al, 1999), lack of p53 leads to organismal rescue and increased fertility, allowing for extra generations with ever-shorter telomere mice (Chin et al, 1999). As first observed in tissue culture, upon telomere shortening, the first barrier to cell proliferation (M1) is imposed by p53/Rb and occurs when telomeres are long enough to allow for further cell divisions (Saretzki et al, 1999). With loss of p53, the ensuing cell proliferation results in complete telomere deprotection, genome instability and cell death during crisis (M2) (Nassour et al, 2019). In this context, loss of cGAS-STING response is consistent with loss of p53 in aging *tert-/-* mutants. However, *tert-/- sting-/-* do not completely phenocopy *tert-/- tp53-/-* mutants. Loss of cGAS-STING does not cause an increase in spontaneous tumor incidence of either WT or *tert-/-* zebrafish. Zebrafish lacking p53 die prematurely, primarily from increased soft tissue tumors (Berghmans et al, 2005). In

contrast, *tert-/- sting-/-* zebrafish lack elevated tumorigenesis characteristic of *tert-/- tp53-/-* mutants (Şerifoğlu et al, 2024). This may be attributed to the downstream consequences of cGAS-STING and type I interferon response. Chronic inflammation may be a key component of increased tumorigenesis in *tp53-/-* mutants. Lack of inflammatory responses may protect *tert-/- sting-/-* zebrafish from early tumorigenesis in the face of increasing DNA damage.

We found that inhibiting cGAS-STING would restore tissue integrity and reduce expression of YAP-TAZ target genes in aging *tert-/-* mutants (El Maï et al, 2023). The YAP-TAZ pathway was recently shown to regulate cGAS-STING in stromal and contractile cells of aging mice (Sladitschek-Martens et al, 2022). Mechanotransduction by YAP-TAZ suppresses the activity of cGAS-STING, preventing senescence in vivo and tissue degeneration of prematurely aging mouse models (Sladitschek-Martens et al, 2022). Our work now shows that cGAS-STING is also required for YAP-TAZ activity upon telomere shortening. This is likely to be the result of an indirect effect on tissue architecture. Restoring tissue integrity would reduce the activation of YAP-TAZ mechanosignaling and modifications to the extracellular matrix.

Loss of cGAS-STING and type I interferon improved the healthspan and the lifespan of aging *tert-/-* mutants. Telomere shortening does not occur simultaneously in all tissues in humans and zebrafish (Aubert et al, 2012; Carneiro et al, 2016). The gut of aging zebrafish presents early dysfunction and telomere shortening. We have recently shown that expressing telomerase specifically in the gut of *tert-/-* mutants prevents telomere shortening and gut dysfunction (El Maï et al, 2023). More importantly, maintaining telomere length in the gut also reverses remote organ dysfunction and longevity of the entire organism. Like *tert-/- sting-/-* zebrafish, gut-specific telomerase expression in *tert-/-* mutants reduces p53 levels and cell senescence in proliferative organs, namely testis and kidney marrow, despite short telomeres and increased levels of γ-H2AX. These lead to increased cell proliferation and restored tissue integrity. We propose that cGAS-STING and type I interferon responses initiated by telomere shortening in specific organs of aging individuals result in systemic chronic inflammation (inflammaging), deteriorating tissue integrity of remote organs by increasing DNA damage and reducing cell proliferation. Thus, inhibition of STING and chronic inflammation in organs primarily affected by telomere shortening, such as the gut and blood, would increase healthspan and lifespan. This provides a new approach for the treatment of telomere biology disorders and to improve healthy aging.

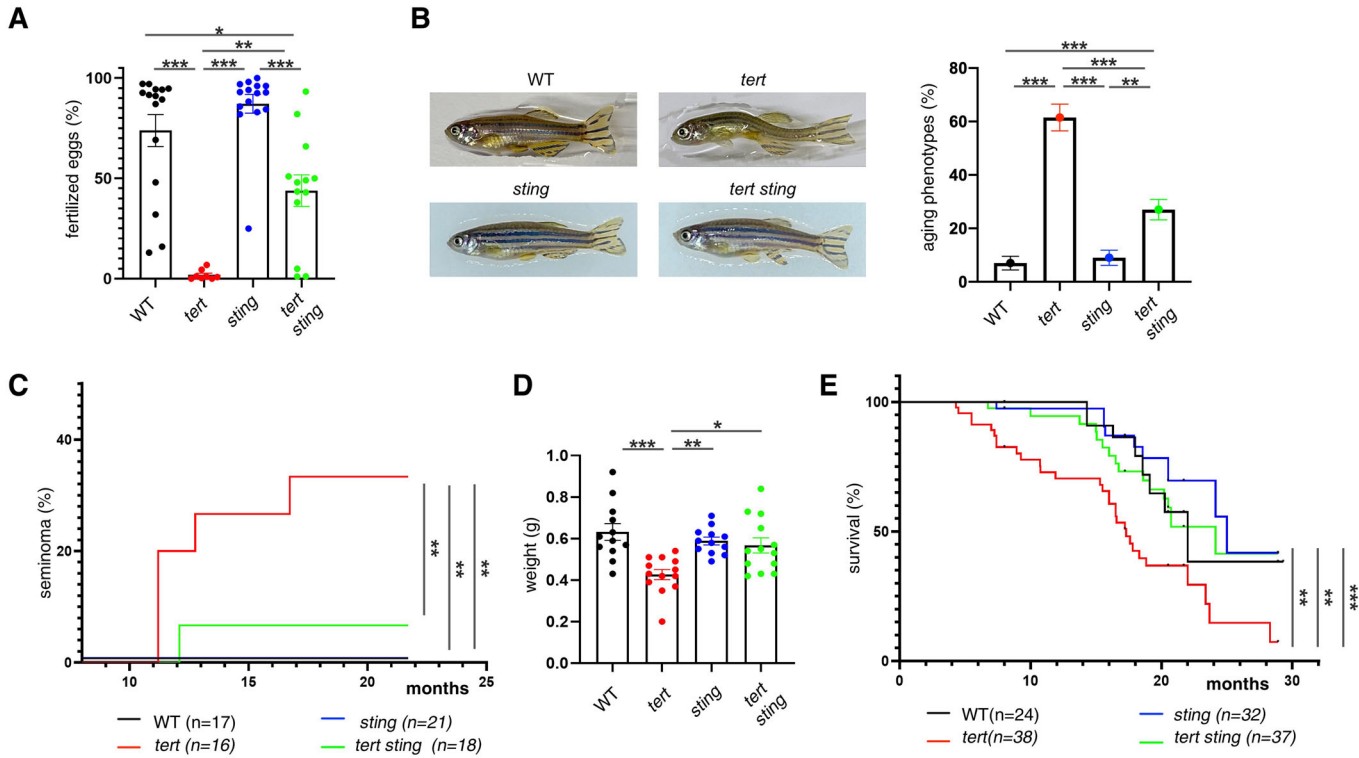

**Figure 6. cGAS-STING pathway rescues premature aging phenotypes.**

(A) Quantification of male fertile capacity ($n_{WT} = 15$, $n_{tert-/-} = 10$, $n_{sting-/-} = 14$, $n_{tert-/-\ sting-/-} = 13$), WT vs *tert-/-* $p = 0.000001$, *sting-/-* vs *tert-/-* $p = 0.000001$, WT vs *tert-/- sting-/-* $p = 0.011$, *sting-/-* vs *tert-/- sting-/-* $p = 0.0001$, *tert-/-* vs *tert-/- sting-/-* $p = 0.002$). (B) Representative images of adult zebrafish and quantification of aging phenotypes scored as kyphosis and cachexia ($n_{WT} = 10$, $n_{tert-/-} = 12$, $n_{sting-/-} = 11$, $n_{tert-/-\ sting-/-} = 12$, WT vs *tert-/-* $p = 0.000001$, *sting-/-* vs *tert-/-* $p = 0.000001$, WT vs *tert-/- sting-/-* $p = 0.001$, *sting-/-* vs *tert-/- sting-/-* $p = 0.003$, *tert-/-* vs *tert-/- sting-/-* $p = 0.000001$). (C) Quantification of seminoma ($n_{WT} = 17$, $n_{tert-/-} = 16$, $n_{sting-/-} = 21$, $n_{tert-/-\ sting-/-} = 18$, WT vs *tert-/-* $p = 0.012$, *sting-/-* vs *tert-/-* $p = 0.002$, *tert-/-* vs *tert-/- sting-/-* $p = 0.027$). (D) Quantification of weight in adult zebrafish ($n_{WT} = 12$, $n_{tert-/-} = 13$, $n_{sting-/-} = 11$, $n_{tert-/-\ sting-/-} = 13$, WT vs *tert-/-* $p = 0.0002$, *sting-/-* vs *tert-/-* $p = 0.004$, *tert-/-* vs *tert-/- sting-/-* $p = 0.012$). (E) Quantification of survival $n_{WT} = 24$, $n_{tert-/-} = 38$, $n_{sting-/-} = 32$, $n_{tert-/-\ sting-/-} = 37$, WT vs *tert-/-* $p = 0.008$, *sting-/-* vs *tert-/-* $p = 0.005$, *tert-/-* vs *tert-/- sting-/-* $p = 0.010$). Data were presented as the mean ± s.e.m. *$p < 0.05$; **$p < 0.01$, ***$p < 0.001$, using a one-way ANOVA and post hoc Tukey test. Seminoma occurrence and survival data were analyzed using log-rank tests, **$p < 0.01$, ***$p < 0.001$.

# Methods

### Reagents and tools table

| Reagent/Resource | Reference or source | Identifier or catalog number |
|---|---|---|
| **Experimental models** | | |
| Zebrafish *tert*<sup>hu3430/hu3430</sup> | Instituto Gulbenkian de Ciência | ZDB-GENO-131125-2 |
| Zebrafish *sting*<sup>sa35634/sa35634</sup> | Instituto Gulbenkian de Ciência | ZDB-ALT-160601-4021 |
| **Antibodies** | | |
| Rabbit anti-p53 (1:1000) | Anaspec | 55342 |
| Rabbit anti-TBK1 (1:1000) | Cell Signaling Technology | 3504 |
| Rabbit anti-pTBK1 (1:1000) | Cell Signaling Technology | 5483 |
| Rabbit anti-IRF3 (1:1000) | Cell Signaling Technology | 11904 |

| Reagent/Resource | Reference or source | Identifier or catalog number |
|---|---|---|
| Rabbit anti-pIRF3 (1:1000) | Cell Signaling Technology | 29047 |
| Rabbit anti-H3 (tri methyl K9) (1:1000) | Abcam | ab8898 |
| Rabbit anti-Actin (1:2000) | Merk | A2066 |
| Mouse anti-Proliferation Cell Nuclear Antigen (PCNA) (1:100) | Santa Cruz | sc56 |
| Rabbit anti-H2A.X (phospho Ser139) (1:100) | Genetex | GTX127342 |
| Goat anti-Rabbit AlexaFluor 488 (1:500) | Invitrogen | A32731 |
| Goat anti-Mouse AlexaFluor 488 (1:500) | Invitrogen | A11001 |
| **Oligonucleotides and other sequence-based reagents** | | |
| qPCR primers | This study | See Materials and Methods |
| **Chemicals, enzymes and other reagents** | | |
| Tissue lysis buffer | Fermentas | K0512 |

| Reagent/Resource | Reference or source | Identifier or catalog number |
| --- | --- | --- |
| Proteinase K | Sigma Aldrich | 3750890 |
| RNase A | Sigma Aldrich | R6513 |
| Chloroform | Sigma Aldrich | C2432 |
| Phenol:chloroform:isoamyl alcohol | Sigma Aldrich | 77617 |
| RSAI | New England Biolabs | R0167 |
| HINFI | New England Biolabs | R0155 |
| [alpha-32P]-dCTP | Revvity | BLU513H250UC |
| MS-222 | Sigma Aldrich | E10521 |
| TrypLE | Gibco | 12604021 |
| Penicillin/Streptomycin | Sigma Aldrich | 4458 |
| Gentamycin | Sigma Aldrich | G1397 |
| Amphotericin b | Sigma Aldrich | A2942 |
| Complete protease and phosphatase inhibitor cocktails | Roche | PPC1010 |
| Nitrocellulose Membrane | Bio-Rad | 1620097 |
| TRIzol | Invitrogen | 15596018 |
| QuantiTect Reverse Transcription kit | Qiagen | 205314 |
| FastStart Universal SYBR Green Master mix | Roche | 4913914001 |
| Sodium citrate buffer | Sigma Aldrich | W302600 |
| PBS | Sigma Aldrich | P4417 |
| DAPI staining | Sigma Aldrich | D9542 |
| DAKO Fluorescence Mounting Medium | Sigma Aldrich | F6182 |
| Nuclear fast red | Sigma Aldrich | N3020 |
| **Software** | | |
| STAR v2.7.10a | | |
| Fiji | https://imagej.net/software/fiji/ | |
| GraphPad Prism 8 | https://www.graphpad.com | |
| **Other** | | |
| StepOne+ Real-time PCR detection system | Applied Biosystems | |
| BioAnalyzer | Agilent 2100 | |
| Leica DM4000 B microscope | Leica Microsystems | |
| Delta Vision Elite | GE Healthcare | |

## Ethics statement

The zebrafish work was conducted according to ARRIVE guidelines and was approved in Portugal by the Ethics Committee of the Instituto Gulbenkian de Ciência and approved by the competent Portuguese authority (Direcção Geral de Alimentação e Veterinária; approval no. 0421/000/000/2015) and in France by the Animal Care Committee of the Institute for Research on Cancer and Aging, Nice, the regional (CIEPAL Côte d'Azur no. 697) and national (French Ministry of Research no. 27673-2020092817202619) authorities.

## Zebrafish lines and maintenance

Zebrafish were maintained in accordance with Institutional and National animal care protocols. To ensure telomere length comparisons and avoid the effects of haploinsufficiency of *tert*+/- heterozygous parental fish, we maintained double heterozygous stock lines (*tert*[AB/hu340] *sting*[AB/sa35631]) as outcrosses to WT AB zebrafish. Experimental fish were obtained by crossing the stock fish. The overall characterization of these four genotypes was performed in F1 sibling animals at 9 months of age. Due to male sex bias in our crosses, that affected mostly *tert*-/- progeny, we were unable to obtain significant numbers of females for analysis and so all of our data except survival analysis are restricted to males.

## Telomere restriction fragment (TRF) analysis by Southern blot

Isolated tissues were first lysed at 50 °C overnight in lysis buffer (Fermentas #K0512) supplemented with 1 mg/ml Proteinase K (Sigma Aldrich) and RNase A (1:100 dilution, Sigma Aldrich). Genomic DNA was then extracted by equilibrated phenol-chloroform (Sigma Aldrich) and chloroform-isoamyl alcohol extraction (Sigma Aldrich). The same amount of gDNA was digested with the RSAI and HINFI enzymes (NEB) for 12 h at 37 °C. After digestion, samples were loaded on a 0.6% agarose gel, in 0.5% TBE buffer, and run on an electrophoresis apparatus (Bio-Rad). The electrophoresis conditions were 110 V for 15 h. Gels were then processed for Southern blotting using a 1.6 kb telomere probe, (TTAGGG)n, labeled with [alpha-32P]-dCTP.

## Fibroblast derival

Nine-month-old zebrafish were sacrificed in 1 g/L of MS-222 (Sigma Aldrich), and the skin was collected in PBS. After three washes in PBS + 2% antibiotics (Penicillin/Streptomycin, Gentamycin and Amphotericin b), the skin was dissociated for 5 min in Tryple (Gibco), cut into small pieces, and let to adhere O/N on coverslip coated by gelatin 2% in presence of few drops of FBS + 2% antibiotics. The day after, when the first fibroblasts were released from the skin, the well was filled with DMEM + 2% antibiotics. After 48 h, fibroblasts were fixed and DAPI staining was performed.

## Western blot

Age and sex matched adult zebrafish were sacrificed in 1 g/L of MS-222 (Sigma Aldrich), and collected tissues (skin, testis, kidney, marrow, and intestine) were immediately snap frozen in liquid nitrogen. Tissues were homogenized in RIPA buffer (sodium chloride 150 mM; Triton X-100 1%, sodium deoxycholate 0.5%, SDS 0.1%, Tris 50 mM, pH = 8.0), including complete protease and phosphatase inhibitor cocktails (Roche diagnostics) with a motor pestle on ice. Homogenized tissues were incubated for 30 min on ice and centrifuged at 4 °C, 13,000 rpm for 10 min. Supernatant was collected and stored at −80 °C until use.

For each sample, 50 µg of protein was loaded per well, separated on 10% SDS-PAGE gels and transferred to Nitrocellulose

Membrane (Bio-Rad #1620097). The membranes were blocked in 5% milk and then incubated with the primary antibody overnight at 4 °C. Antibody complexes were visualized by enhanced chemiluminescence (ECL) after incubation with the appropriate HRP-conjugated secondary antibody. Antibodies concentrations: anti-p53 (1:1000, Anaspec, 55342), anti TBK1 (1:1000, CST, 3504), anti pTBK1 (1:1000, CST, 5483), anti-IRF3 (1:1000, CST, 11904), anti pIRF3 (1:1000, CST, 29047), anti-Actin (1:2000, Merk, A2066), anti-H3(tri methyl K9) (1:1000, Abcam, ab8898).

## Real-time quantitative PCR

Age and sex matched adult zebrafish were sacrificed in 1 g/L of MS-222 (Sigma Aldrich), and collected tissues (skin, testis, kidney, marrow and intestine) were immediately snap frozen in liquid nitrogen. RNA extractions were performed in TRIzol (Invitrogen) by mashing tissues with a motorized pestle in a 1.5 mL Eppendorf tube. After incubation at room temperature (RT) for 10 min, TRIzol, chloroform extractions were performed. The quality of RNA samples was assessed through the BioAnalyzer (Agilent 2100). Retro-transcription into cDNA was performed using QuantiTect Reverse Transcription kit (Qiagen).

Quantitative PCR (qPCR) was performed using FastStart Universal SYBR Green Master mix (Roche) and a StepOne+ Real-time PCR Detection System (Applied Biosystems). qPCRs were carried out in duplicate for each cDNA sample. Relative mRNA expression was normalized against *rps11* mRNA expression using the $2^{-dCT}$ method.

|  | Gene ID: | Forward primer | Reverse primer |
|---|---|---|---|
| *isg15* | ZDB-GENE-021211-1 | ACTCGGTGACGATGCAGC | TGGGCACGTTGA AGTACTGA |
| *ifn-i* | ZDB-GENE-030721-3 | CAAGATACGCAAAGC CAGCA | GTGGCTTTTCAC AACTCTCC |
| *cdkn 2a/b* | ZDB-GENE-081104-306 | GAGGATGAACTGACCA CAGCA | CAAGAGCCAAAG GTGCGTTAC |
| *cdkn1a* | ZDB-GENE-070705-7 | CAGCGGGTTTACA GTTTCAGC | TGAACGTAGGAT CCGCTTGT |
| *il1b* | ZDB-GENE-040702-2 | CGCTCCACATCTC GTACTCA | ATACGCGGTGCT GATAAACC |
| *tgfb1b* | ZDB-GENE-091028-1 | ACCCCAGTTCAGCAC ACCATAG | TCGAAACTCGGC CTGGTAGA |
| *mmp15a* | ZDB-GENE-070817-4 | GGGTCATGCTCTGGG GGTTGG | AGTGGTGACAGTC TCTGGAGATCCA |
| *ctgf* | ZDB-GENE-030131-102 | ACTCCCCTCGTCAAA ACACC | GGGACCGTATGT CTCCTCCT |
| *cyr61* | ZDB-GENE-040426-3 | CCGTGTCCACATGTA CATGGG | GGTGCATGAAAGA AGCTCGTC |
| *mavs* | ZDB-GENE-070112-1402 | AGTAGAAGCCGCGAGA GGTA | GGCTTCGATCTC TTCCCGAT |
| *rps11* | ZDB-GENE-040426-2701 | ACAGAAATGCCCCTT CACTG | GCCTCTTCTCAA AACGGTTG |

## Histology

Age and sex-matched adult zebrafish were sacrificed in 1 g/L of MS-222 (Sigma Aldrich), fixed for 72 h in 4% paraformaldehyde and

decalcified in 0.5 M EDTA for 48 h at room temperature. Whole fish were then paraffin-embedded to perform 5 µm sagittal section slides. Slides were stained with hematoxylin (Sigma Aldrich) and eosin (Sigma Aldrich) for histopathological analysis. Microphotographs ($N \geq 6$ fish per genotype) were acquired in a Leica DM4000 B microscope coupled to a Leica DFC425 C camera (Leica Microsystems).

## Immunofluorescence

Deparaffinized and rehydrated slides were microwaved for 20 min at 550 W in citrate buffer (10 mM Sodium Citrate, pH 6.0) for antigen retrieval. Slides were washed twice in PBS for 5 min and blocked for 1 h at RT in 0.5% Triton, 1% DMSO, 5% normal goat serum in PBS (blocking solution). Subsequently, slides were incubated overnight at 4 °C with a 1:50 dilution of primary antibody in blocking solution. The following primary antibodies were used: mouse monoclonal antibody against Proliferation Cell Nuclear Antigen (PCNA), sc56, (Santa Cruz), rabbit polyclonal Histone H2A.XS139ph (phospho Ser139) GTX127342 (Genetex). Following two PBS washes, overnight incubation at 4 °C was performed 1:500 dilution of goat anti-rabbit secondary antibody AlexaFluor 488 (Invitrogen) and goat anti-mouse secondary antibody AlexaFluor 488 (Invitrogen). Finally, after DAPI staining (Sigma Aldrich), slides were mounted DAKO Fluorescence Mounting Medium (Sigma Aldrich).

Immunofluorescence images were acquired on Delta Vision Elite (GE Healthcare) using an OLYMPUS 60x/1.42 objective. For quantitative and comparative imaging, equivalent acquisition parameters were used. The percentage of positive nuclei was determined by counting a total of 150–1000 cells per slide, depending on the tissue ($N \geq 6$ zebrafish per genotype).

## Senescence associated beta galactosidase staining

Age and sex matched 9-month-old zebrafish were sacrificed in 1 g/l of MS-222 (Sigma Aldrich, MO, USA), fixed in 4% paraformaldehyde in PBS for 72 h at 4 °C, washed three times for 1 h in 1x PBS pH 7.4 and 1 h in 1x PBS pH 6.0 at 4 °C. Beta Galactosidase staining was performed for 24 h at 37 °C in 5 mM potassium ferrocyanide, 5 mM potassium ferricyanide, 2 mM MgCl$_2$, and 1 mg/mL X-Gal, in 1x PBS pH 6.0. After staining, fish were washed three times for 5 min in 1x PBS pH 7.0, and processed for decalcification and paraffin embedding. Paraffin blocks were sectioned sagittally, 5 µm in thickness and co-stained with nuclear fast red (Sigma Aldrich).

## Fertility assays

In order to assess male fertility, 9-month-old males from the four different genotypes were separately housed overnight in external breeding tanks with a single young (3–6 months old) WT female. Breeding pairs were left to cross and lay eggs in the following morning and embryos were collected ~4 h post fertilization (hpf) and allowed to develop at 28 °C. Assessment of fertilized eggs and embryo viability was conducted between 4 and 6 hpf. At least 12 independent crosses were conducted for each genotype to evaluate male fertility. Only successful breeding trials, defined as events in which a clutch of eggs was laid by a female, were scored.

## Fixation for histology and tumor evaluation

Fish were screened weekly for the presence of macroscopic tumors (Berghmans et al, 2005). Zebrafish were euthanized with 1 g/L of MS-222 (Sigma, MO, USA), followed by fixation in 10% neutral buffered formalin for 72 h and decalcified in 0.5 M EDTA for 48 h at room temperature. Samples were then paraffin-embedded to perform 5 μm sagittal section slides. Slides were stained with hematoxylin and eosin and assessed for the presence of tumors.

## RNA-seq analysis of TE families

Raw FASTQ read files obtained from the SRA (PRJNA937311) were aligned to the reference zebrafish genome (GRCz10/danRer10) using STAR v2.7.10a with the following parameters to retain multi-mapping reads: --outMultimapperOrder Random --outSAMmultNmax 1 --outFilterMismatchNmax 3 --winAnchorMultimapNmax 100 --outFilterMultimapNmax 100 --alignSJDBoverhangMin 1. Expression of transposable element families was quantified using TEtranscripts v2.2.393 with parameter –mode=multi to estimate transposable element abundances from multimapped alignments using the pre-generated GENCODE danRer10 TE GTF file from the Hammel lab FTP site (https://labshare.cshl.edu/shares/mhammelllab/www-data/TEtranscripts/TE_GTF). Counts were normalized to counts per million (CPM) and differential expression was assessed in R v3.1.4 (R Core Team, 2022) on the combined transposable element /gene counts using the edgeR package v3.34.189 exact test.

## Statistical analysis

Graphs and statistical analyses were performed in GraphPad Prism 8 software, using one-way ANOVA test, Tukey's post-correction or unpaired $t$-test. A critical value for significance of $p < 0.05$ was used throughout the study. For survival analysis, log-rank tests were performed using GraphPad Prism 8 to determine statistical differences in survival curves.

# Data availability

All data generated or analyzed during this study are included in this published article and its extended information files. RNAseq data are available to the sequence read archive (SRA) database under accession no. PRJNA937311. Otherwise, no large-scale data sets amenable to public repository deposition were generated in this study.

The source data of this paper are collected in the following database record: biostudies:S-SCDT-10_1038-S44318-025-00482-5.

# Peer review information

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

## Acknowledgements
We are grateful to our team members for fruitful discussions and advice. Especially, we thank Dr. Hervé Técher for critically reading the manuscript. This work was supported by the Université Côte d'Azur - Académie 4 (Installation Grant: Action 2 - 2019), Agence Nationale de la Recherche (ANR-21-CE14-0054) and Institut National du Cancer (INCa, PLBIO21-228). N.S. was supported by a PhD fellowship from La Ligue Contre le Cancer. ET and PB thank the German Research Foundation, grant nos. 322977937/GRK2344. We are grateful to the PEMAV fish facility, Imaging core facility (PICMI) and the Genomics facilities at the IRCAN, supported by FEDER, Région Provence Alpes-Côte d'Azur, Conseil Départemental 06, ITMO Cancer Aviesan (plan cancer), Cancéropole Provence Alpes-Côte d'Azur, Gis Ibisa, CNRS and Inserm. This work has been supported by the French government, through the France 2030 investment plan managed by the Agence Nationale de la Recherche, as part of the Université Côte d'Azur's Initiative of Excellence, reference ANR-15-IDEX-01. The funders had no role in study design, data collection and analysis, decision to publish or preparation of the manuscript.

## Author contributions
**Naz Şerifoğlu**: Data curation; Formal analysis; Investigation; Methodology; Writing—original draft. **Giulia Allavena**: Data curation; Formal analysis; Writing—review and editing. **Bruno Lopes-Bastos**: Investigation; Methodology; Writing—review and editing. **Marta Marzullo**: Formal analysis; Investigation. **Andreia Marques**: Investigation; Methodology. **Pauline Colibert**: Methodology. **Pavlos Bousounis**: Visualization; Methodology. **Eirini Trompouki**: Project administration; Writing—review and editing. **Miguel Godinho Ferreira**: Conceptualization; Supervision; Funding acquisition; Writing—original draft; Project administration; Writing—review and editing.

Source data underlying the figure panels in this paper may have individual authorship assigned. Where available, figure panel/source data authorship is listed in the following database record: biostudies:S-SCDT-10_1038-S44318-025-00482-5.

## Disclosure and competing interests statement
The authors declare no competing interests.

