## [Peer Review File · The EMBO Journal]

cGAS-STING are responsible for premature aging of telomerase-deficient zebrafish

Naz Şerifoğlu, Giulia Allavena, Bruno Bastos-Lopes, Marta Marzullo, Andreia Marques, Pauline Colibert, Pavlos Bousounis, Eirini Trompouki, and Miguel Ferreira

Corresponding author: Miguel Ferreira (Miguel-Godinho.FERREIRA@unice.fr)

Review Timeline:

Submission Date:	9th Apr 24
Editorial Decision:	10th May 24
Revision Received:	14th Nov 24
Editorial Decision:	3rd Apr 25
Revision Received:	16th Apr 25
Accepted:	22nd May 25

Editor: Daniel Klimmeck

Transaction Report:

Dear Miguel,

Thank you for submitting your manuscript for consideration by the EMBO Journal, as well as for your patience with our feedback at this time of the year. Your work has now been seen by two referees with expertise in longevity signaling and immunology and we received comments from both of them which are shown below.

Given the overall interest stated and broader angle of your findings, we are able to invite you to revise your manuscript experimentally to address the referees' comments. I need to stress though that we do require strong support from the referees on a revised version of the study in order to move on to publication of the work.

I would appreciate if you could contact me during the next weeks for exchange e.g. a video call to discuss your perspective on the comments and potential plan for revisions.

Please feel free to also contact me if you have any questions or need further input on the referee comments.

When submitting your revised manuscript, please carefully review the instructions below.

Please feel free to approach me any time should you have additional questions related to this.

Thank you for the opportunity to consider your work for publication.

I look forward to your revision.

Best regards,

Daniel

Daniel Klimmeck, PhD
Senior Editor
The EMBO Journal

Instruction for the preparation of your revised manuscript:

- 1) a .docx formatted version of the manuscript text (including legends for main figures, EV figures and tables). Please make sure that the changes are highlighted to be clearly visible.
- 2) individual production quality figure files as .eps, .tif, .jpg (one file per figure).
- 3) a .docx formatted letter INCLUDING the reviewers' reports and your detailed point-by-point response to their comments. As part of the EMBO Press transparent editorial process, the point-by-point response is part of the Review Process File (RPF), which will be published alongside your paper.
- 4) a complete author checklist, which you can download from our author guidelines ([https://wol-prod-cdn.literatumonline.com/pb-assets/embo-site/Author Checklist%20-%20EMBO%20J-1561436015657.xlsx](https://wol-prod-cdn.literatumonline.com/pb-assets/embo-site/Author%20Checklist%20-%20EMBO%20J-1561436015657.xlsx)). Please insert information in the checklist that is also reflected in the manuscript. The completed author checklist will also be part of the RPF.
- 5) Please note that all corresponding authors are required to supply an ORCID ID for their name upon submission of a revised manuscript.
- 6) It is mandatory to include a 'Data Availability' section after the Materials and Methods. Before submitting your revision, primary datasets produced in this study need to be deposited in an appropriate public database, and the accession numbers and database listed under 'Data Availability'. Please remember to provide a reviewer password if the datasets are not yet public (see <https://www.embopress.org/page/journal/14602075/authorguide#datadeposition>). In case you have no data that requires deposition in a public database, please state so in this section. Note that the Data

Availability Section is restricted to new primary data that are part of this study.

7) Our journal encourages inclusion of *data citations in the reference list* to directly cite datasets that were re-used and obtained from public databases. Data citations in the article text are distinct from normal bibliographical citations and should directly link to the database records from which the data can be accessed. In the main text, data citations are formatted as follows: "Data ref: Smith et al, 2001" or "Data ref: NCBI Sequence Read Archive PRJNA342805, 2017". In the Reference list, data citations must be labeled with "[DATASET]". A data reference must provide the database name, accession number/identifiers and a resolvable link to the landing page from which the data can be accessed at the end of the reference. Further instructions are available at .

8) At EMBO Press we ask authors to provide source data for the main and EV figures. Our source data coordinator will contact you to discuss which figure panels we would need source data for and will also provide you with helpful tips on how to upload and organize the files.

Numerical data can be provided as individual .xls or .csv files (including a tab describing the data). For 'blots' or microscopy, uncropped images should be submitted (using a zip archive or a single pdf per main figure if multiple images need to be supplied for one panel). Additional information on source data and instruction on how to label the files are available at .

9) We replaced Supplementary Information with Expanded View (EV) Figures and Tables that are collapsible/expandable online (see examples in <https://www.embopress.org/doi/10.15252/emj.201695874>). A maximum of 5 EV Figures can be typeset. EV Figures should be cited as 'Figure EV1, Figure EV2' etc. in the text and their respective legends should be included in the main text after the legends of regular figures.

11) For data quantification: please specify the name of the statistical test used to generate error bars and P values, the number (n) of independent experiments (specify technical or biological replicates) underlying each data point and the test used to calculate p-values in each figure legend. The figure legends should contain a basic description of n, P and the test applied. Graphs must include a description of the bars and the error bars (s.d., s.e.m.).

We realize that it is difficult to revise to a specific deadline. In the interest of protecting the conceptual advance provided by the work, we recommend a revision within 3 months (8th Aug 2024). Please discuss the revision progress ahead of this time with the editor if you require more time to complete the revisions.

Referee #1:

The manuscript by Şerifoğlu et al. investigates how cGAS-STING mediates the impact of telomere deficiency on cellular and organismal aging. By drawing a direct mechanistic connection between telomere dysfunction and sterile inflammation, i.e., two major effector of the aging process, this work is novel, of broad significance and highly relevant for the aging field. I congratulate the authors for having produced a rigorous, clear, and convincing work.

The authors adopt tert deficient zebrafish as their main experimental model system. The results here presented build up on previous results from this team, where they developed a tert ^{-/-} zebrafish line that recapitulates several molecular, cellular, organ, and organismal hallmarks of aging. In this work, the authors demonstrate that several pro-aging phenotypes featured in their tert ^{-/-} zebrafish line are mediated by the cGAS STING immune pathway. Remarkably, the authors show that cGAS-STING signaling is required for p53 expression after telomere-shortening-dependent DNA damage response. From tissue proliferation to cellular senescence in vivo, to organismal lifespan, cGAS STING appears to mediate virtually all the downstream adverse effects of telomerase dysfunction. Even more, sting mutants appear to even outlive wt zebrafish (Figure 5e). Overall, the results are impressive and clearly bring home the concept that cGAS STING is a key downstream effector for telomerase-dysfunction dependent DNA damage response.

My only question to this study is whether there are adverse phenotypes in the tert^{-/-} line that are NOT mediated by cGAS. I have no further comments and I congratulate the authors for a well conducted work.

Referee #2:

The manuscript aims to address the role of telomere shortening in activating the cGAS-STING pathway and its downstream effects on type I interferon response, inflammation, aging, and tissue integrity in zebrafish. The manuscript employs a multifaceted experimental strategy, combining genetic mutations, telomere length assessments, cellular analyses, and whole-organism studies. Given the complexity of these processes, the integration of multiple analyses, including gene expression, telomere length assessment, and cellular and tissue-level studies, provides a broad view of the topic. The manuscript is well written and structured and the findings are relevant and of interest to the proposed research field. However, the scope of this manuscript is substantial, which may lead to a dilution of specific key findings. The manuscript suggests a mechanism by which telomere shortening activates the cGAS-STING pathway, but it lacks concrete evidence to support these hypotheses. Further experimental validation of these mechanisms would be valuable. More specific comments are shown below:

The data suggest that the cGAS-STING pathway is activated in response to telomere shortening, but the manuscript does not provide direct evidence of this activation, nor does it explain why telomerase-deficient fish show an upregulation of transposable elements (TEs). The authors should consider including additional experiments that demonstrate a functional link between telomere attrition and cGAS-STING activation.

Is the transcriptional activation of L1 retrotransposable elements specific to telomere shortening? Is it also observed in naturally aged wild-type zebrafish? What is the functional relevance of L1 retrotransposable elements to micronuclei?

The overexpression of MAVS mRNA appears heterogeneous i.e., present in some tissues but not in others. Does this indicate that the inflammatory response is also heterogeneous? Are some tissues more vulnerable to increased RNA sensing or interferon (IFN) response than others?

The data suggest that p53 levels in the tert^{-/-} sting^{-/-} mutants are similar to those in wild-type and sting^{-/-} siblings. The authors suggest that this similarity is likely due to a reduction in IFNs. Does this hold true for RNA sensing through pathways like ZBP1-MAVS in MAVS mutant fish?

The authors should include a few more senescence-associated secretory phenotype (SASP) markers before concluding that the SASP response is dampened in STING^{-/-} fish.

The authors should clarify whether STING abrogation directly inhibits senescence and leads to increased cell proliferation (through an otherwise unknown mechanism) or if it does so indirectly, for example, by dampening IFN-mediated inflammation. If the latter is true, it would suggest that inhibiting IFN could produce similar results in tert-deficient zebrafish.

The data suggest that telomere attrition (and by inference DNA damage) leads primarily to the activation of the interferon (IFN) response. This is supported by evidence that STING abrogation rescues several age-related features and reduces cellular senescence. However, other research indicates that multiple pro-inflammatory pathways are activated in response to genome instability, which also leads to cellular senescence and premature aging. How do the authors reconcile these findings with their own, which suggest that DDR and senescence triggered by short telomeres only require an active cGAS-STING pathway?

Referee

#1:

The manuscript by Şerifoğlu et al. investigates how cGAS-STING mediates the impact of telomere deficiency on cellular and organismal aging. By drawing a direct mechanistic connection between telomere dysfunction and sterile inflammation, i.e., two major effector of the aging process, this work is novel, of broad significance and highly relevant for the aging field. I congratulate the authors for having produced a rigorous, clear, and convincing work.

The authors adopt *tert* deficient zebrafish as their main experimental model system. The results here presented build up on previous results from this team, where they developed a *tert* $-/-$ zebrafish line that recapitulates several molecular, cellular, organ, and organismal hallmarks of aging. In this work, the authors demonstrate that several pro-aging phenotypes featured in their *tert* $-/-$ zebrafish line are mediated by the cGAS-STING immune pathway. Remarkably, the authors show that cGAS-STING signaling is required for p53 expression after telomere-shortening-dependent DNA damage response. From tissue proliferation to cellular senescence *in vivo*, to organismal lifespan, cGAS-STING appears to mediate virtually all the downstream adverse effects of telomerase dysfunction. Even more, *sting* mutants appear to even outlive wt zebrafish (Figure 5e). Overall, the results are impressive and clearly bring home the concept that cGAS-STING is a key downstream effector for telomerase-dysfunction dependent DNA damage response.

My only question to this study is whether there are adverse phenotypes in the *tert* $-/-$ line that are NOT mediated by cGAS. no further comments and I congratulate the authors for a well conducted work.

We thank Referee #1 for the encouraging review. We believe that the cGAS-STING pathway does not completely inhibit the adverse effects of shorter telomeres and is a time-dependent rescue for certain phenotypes. Indeed, at 9 months of age absence of *sting* does not restrain DNA damage markers, such as γ H2AX and micronuclei (Fig. 1E and 3A-E). Moreover, it only partially rescues the aging phenotypes (Fig 6B), the reduction of mature sperm area (Fig. 5H), and it does not completely recover male fertility. Indeed, by the age of 13 months, *tert sting* double mutant fish become sterile (see Supplementary Fig. 5 below, text page 9 - line 256-257), although their lifespan is not

Supplementary figure 5: male fertility

compromised.

Our interpretation for these results is based on the two sequential stages of telomere shortening. In the first, when most telomeres are long enough to support further cell divisions but nonetheless trigger DNA Damage Responses and stabilize p53 levels (generally known as M1 Hayflick limit or replicative senescence). And a second stage, when telomeres are too short to protect chromosomes from end-joining DNA repair and cause genome instability (also known as M2 or crisis). Our work suggests that the first stage initiates cGAS-STING IFN activation and that this leads to systemic defects caused primarily by inflammatory responses. In absence of cGAS-STING, this first stage would not be greatly impacted by genome instability, as telomeres would still sustain further cell divisions (similar to inhibition of *tp53* in primary cell cultures). However, once telomeres shorten to critical

length, they are unable to sustain further cell divisions and cause tissue collapse. Cell culture studies from the Karlseder lab using human primary cells (inhibited for *tp53* function) have shown that cGAS-STING is also activated during crisis and that absence of *sting* function would allow for further cell divisions albeit under genome instability. Although this has not yet been studied in mouse models, it is expected that crisis would impair tissue homeostasis leading to loss of organism viability, similar to what we observed in our double mutant at later stages. So, cGAS-STING causes age-associated phenotypes in early telomere dysfunction but is unable to rescue the consequences of loss of telomere protection at later stages.

Referee #2:

The manuscript aims to address the role of telomere shortening in activating the cGAS-STING pathway and its downstream effects on type I interferon response, inflammation, aging, and tissue integrity in zebrafish.

The manuscript employs a multifaceted experimental strategy, combining genetic mutations, telomere length assessments, cellular analyses, and whole-organism studies. Given the complexity of these processes, the integration of multiple analyses, including gene expression, telomere length assessment, and cellular and tissue-level studies, provides a broad view of the topic. The manuscript is well written and structured and the findings are relevant and of interest to the proposed research field. However, the scope of this manuscript is substantial, which may lead to a dilution of specific key findings. The manuscript suggests a mechanism by which telomere shortening activates the cGAS-STING pathway, but it lacks concrete evidence to support these hypotheses. Further experimental validation of these mechanisms would be valuable.

More specific comments are shown below:

1 - The data suggest that the cGAS-STING pathway is activated in response to telomere shortening, but the manuscript does not provide direct evidence of this activation, nor does it explain why telomerase-deficient fish show an upregulation of transposable elements (TEs). The authors should consider including additional experiments that demonstrate a functional link between telomere attrition and cGAS-STING activation.

We thank Referee #2 for this recommendation. We believe that the activation of cGAS-STING pathway is due to concomitant series of events, occurring during telomere shortening, and already reported from previous research. Although we do not have directed evidence of increased cytoplasmic DNA, our previous study (El Mai et al., 2020) showed increased mitochondria membrane rupture in *tert*^{-/-} and in the present work we show a general mobilization of transposable elements. Both these events will likely lead to cytoplasmic DNA that may lead to cGAS-STING activation.

Recent research from other groups shows that critically short telomeres are linked to genome instability through TE derepression. Using late-generation telomerase knockout (*Terc*^{-/-}) mice, Zhao and colleagues show that critically short telomeres modulated the expression of retrotransposons at subtelomeric regions, promoting genomic mutations, deletion or amplification, chromatin interactions, changing chromatin accessibility and H3K9me3 profiling (PMID: 37130870).

In our analysis, we observe a general mobilization of transposon spanning different organs and we now tested specifically the methylation status of Histone 3 in the skin. Consistently, H3K9me3 is significantly downregulated and, associated to this downregulation, we observe an increase in LTR2 element, confirming transposon de-repression (new Fig. 1F and G, below; text added in the results section: page 4-5, line 104-114).

In order to evaluate the contribution of mitochondrial dysfunction on cGAS-STING activation, we used second generation (G2) *tert*^{-/-} fish. G2 *tert*^{-/-} larvae recapitulate the main features of G1 *tert*^{-/-} fish in their short lifespan (PMID: 23349637, 23744274, 32554492 and 32427102), thus allowing us to perform faster experiments compatible with the manuscript revision period. Consistent with adult G1 *tert*^{-/-} fish, 5 days post fertilization (5dpf) G2 *tert*^{-/-} have very short telomeres, and an increased expression of interferon-stimulated genes associated with an increase in senescence and inflammatory markers (new Supplementary Fig. 2, text in page 5 - line 132-137, page 8 – line 208-210, page 10 – line 277-278). We tested if blocking mitochondrial DNA release to the cytoplasm using VBIT-4 (inhibitor of VDAC1 oligomerization), was able to restrain interferon response and senescence in G2 *tert*^{-/-} larvae. Our preliminary results (below) show that while the administration of 2uM of VBIT-4 reduces expression of *p16* and *p21* markers of senescence, it is still uncertain the effect on SASP and inflammatory markers (*il1b* and *tgf1b*), while it seems not to affect interferon response (*isg15*). More experiments using higher dosages of VBIT-4 will better address this point.

In conclusion, we anticipate that micronuclei, chromatin rearrangements and transposon mobilization are primary factors that lead to cGAS-STING activation upon telomere erosion. Similarly to what was recently observed in human cells (PMID: 39191740), mitochondrial dysfunction might be the amplifier of senescence and inflammatory signaling. To thoroughly address this point, we plan to perform more experiments using different inhibitors of mitochondrial DNA release in the cytosol and directly sequence cGAS-bound cytoplasmic nucleic acids in *tert*^{-/-} zebrafish.

2 - Is the transcriptional activation of L1 retrotransposable elements specific to telomere shortening? Is it also observed in naturally aged wild-type zebrafish? What is the functional relevance of L1 retrotransposable elements to micronuclei?

In the work by Zhao and colleagues (PMID: 37130870), the authors fail to observe a specific activation of L1 retroelements upon telomere shortening in mice. In zebrafish, our RNAseq analysis revealed a substantial mobilization of LTRs and DNA transposons rather than L1 retroelements. In contrast to mammals, zebrafish LINE retrotransposons account only for 4% of total genome (PMID: 34987056) and, thus, unlikely to be highly the most representative TEs, unless specifically de-repressed. To address Referee #2's question directly, we performed RT-qPCR for L1 retroelements in

tissues where we observed a reduction of H3K9me3 and upregulation of LTR2. Consistent with the RNAseq data, we failed to observe a significant expression of L1-5 (below). In conclusion, the major TE contributors to activation cGAS-STING in zebrafish are likely to be LTR elements.

3 - The overexpression of MAVS mRNA appears heterogeneous i.e., present in some tissues but not in others. Does this indicate that the inflammatory response is also heterogeneous? Are some tissues more vulnerable to increased RNA sensing or interferon (IFN) response than others?

According to our results, RNA sensing genes (*mavs*, *rigl*, *mda5*) are less expressed when compared to interferon response genes (*ifn1*, *isg15*), showing the lowest expression in the gut. This suggests that the RNA sensing pathway is variable and probably not the main contributor to the IFN response. As Referee #2 points out, *mavs* expression appears to be overall variable and only significantly increased in testis of *tert*^{-/-}, but not in the gut or kidney marrow. Similarly, *rigl* and *mda5* show no significant difference in both gut and testis and *rigl* (but not *mavs*) show a significant increase in kidney marrow. Our results suggest that, in face of clear cGAS-STING activation, the RNA sensor pathway appears to take a secondary role, perhaps revealing a later response, similar to the ZBP1/RIG-I activation observed by the Karlseder lab in response to TERRA, the telomeric RNA transcript expressed in human cells undergoing crisis.

To confirm a primary role for *sting* in IFN expression, we included RT-qPCRs profiles for the RNA sensing genes in *sting*^{-/-} and *tert*^{-/-} *sting*^{-/-} double mutants (new Supplementary Fig. 3). Similar to *tert*^{-/-}, *tert* *sting* double mutants do not show a statistically significant increase in *mavs*, *rigl*, *mda5* levels. Finally, we tried to evaluate the response heterogeneity for the RNA sensing genes using the coefficient of variation (CV). However, we were unable to detect an increased variability in gene expression of *mavs*, *rigl*, *mda5* when compared to *ifn1* and *isg15*.

4 - The data suggest that p53 levels in the *tert*^{-/-} *sting*^{-/-} mutants are similar to those in wild-type and *sting*^{-/-} siblings. The authors suggest that this similarity is likely due to a reduction in IFNs. Does this hold true for RNA sensing through pathways like ZBP1-MAVS in MAVS mutant fish?

Expression data of *mavs* is only significantly reduced in the gut of *sting*^{-/-} and *tert*^{-/-} *sting*^{-/-} fish, while no difference is evident in other tissues (Supplementary Fig 3). Given the consistency of p53 levels on the analyzed tissues, we anticipate that reduction in p53 is unlikely to be a consequence of the expression/activation of the RNA sensing pathway.

To address Referee #2's point directly, we performed morpholino (MO) gene downregulation experiments in G2 *tert*^{-/-} zebrafish (data added to Supplementary Fig.2F-H). Injecting *mavs* MO at the one-cell stage of G2 *tert*^{-/-} fish, we significantly reduce *mavs* expression compared to scrambled MO controls. However, neither senescence markers (p16 and p21) nor p53 levels were reduced in G2 *tert*^{-/-} in response to lower *mavs* expression (new Supplementary Fig. 3F-H, below. Text added to the results section: page 7 - lines 183-187 and page 8 - lines 208-210). Thus, in contrast to *sting* mutants,

our results suggest that *mavs* expression is not required for higher p53 levels upon telomere shortening.

5 - The authors should include a few more senescence-associated secretory phenotype (SASP) markers before concluding that the SASP response is dampened in STING^{-/-} fish.

Following Referee #2's suggestion, we investigated additional SASP markers according to the SenNet recommendations published this year (PMID: 38831121), which propose to evaluate IL1a, IL1b, IL6, and *serpine1*, even though tissue-specific SASPs exist. Following these recommendations, we considered the expression of IL1a. However, similar to amphibians, fish possess only one copy of IL1, that better resembles the mammalian IL1b (PMID: 37417734). We have already provided evidence for expression IL1b in our manuscript (Fig. 4B-E). In addition, we examined *serpine1* and *il6* expression as SASP markers. We observed no significant increase in *serpine1* expression level in our prematurely aged fish (below). Unfortunately, we do not yet have results for *il6* expression. Nevertheless, we agree with Referee #2 on the importance of using further SASP markers in our studies. To strengthen our understanding of SASP in zebrafish, we are setting up RNAseq analysis on different tissues to investigate additional SASP markers. Unfortunately, these experiments will take longer than the time required for revisions of our manuscript.

6 -The authors should clarify whether STING abrogation directly inhibits senescence and leads to increased cell proliferation (through an otherwise unknown mechanism) or if it does so indirectly, for example, by dampening IFN-mediated inflammation. If the latter is true, it would suggest that inhibiting IFN could produce similar results in *tert*-deficient zebrafish.

Our work suggests that dampening the interferon response (through *sting* abrogation) inhibits senescence and leads to increased cell proliferation. To support this idea, recent human cell culture studies report that prolonged treatment with interferon beta alone generates genome instability and cell senescence (PMID: 16436515, 38926338). Moreover, upon telomere damage in mice, the interferon beta is required for amplification of DNA damage response and the induction of cellular senescence (PMID: 25921537). In this study, ablation of interferon receptor, *ifnar1*, rescued telomere damage associated senescence and extended the lifespan of *Terc*^{-/-} mice.

We nevertheless attempted to address this point directly by treating zebrafish larvae with 300 U/ml and 1000 U/ml of human interferon beta (IFN β). However, we were unable to produce an interferon response and increased senescence in our model (below). This outcome is likely the result of structural differences between human and zebrafish interferon beta. In order to circumvent this problem, we are planning additional experiments that abrogate the IFN response using mutants of the zebrafish interferon receptor.

7 - The data suggest that telomere attrition (and by inference DNA damage) leads primarily to the activation of the interferon (IFN) response. This is supported by evidence that STING abrogation rescues several age-related features and reduces cellular senescence. However, other research indicates that multiple pro-inflammatory pathways are activated in response to genome instability, which also leads to cellular senescence and premature aging. How do the authors reconcile these findings with their own, which suggest that DDR and senescence triggered by short telomeres only require an active cGAS-STING pathway?

We thank Referee #2 for this comment. We agree that genome instability can indeed activate a range of pro-inflammatory pathways, many of which contribute to cellular senescence and premature aging. However, we believe that the cGAS-STING pathway plays a major role in our model of telomere attrition. To the best of our knowledge, the main pathways activated in response to genome instability are the DNA and RNA sensor pathways. Namely, the cGAS-STING pathway, RIGI, MDA5 and MAVS pathway, and the Toll-like receptors (TLR3, TLR7, and TLR9). Among them, we primarily investigated the cGAS-STING pathway. cGAS-STING is strongly activated in our model and removal of *sting* significantly reduces senescence and other age-associated phenotypes. In addition, we examined the role of the RNA sensors *rig1*, *mda5* and *mavs*. These genes exhibit only mild transcriptional upregulation in response to telomere attrition and, we failed to observe a significant effect on senescence in absence of *mavs* (see point 4 above), attesting their secondary role in our model of premature aging. With regards to Toll-like receptors, TLR3 and TLR7 recognize self-RNA, while TLR9 senses self-dsDNA. These receptors are primarily expressed in immune cells and their expression declines with age (PMID: 12391175). Although we did not investigate the role of TLRs in our current study, our previous RNAseq analysis of gut, testis and kidney marrow tissues (PMID: 37142828), did not reveal a significant variation in expression of TLRs.

Thus, our results suggest that cGAS-STING is the main sensor of telomere attrition and ensuing genome instability in telomerase mutant zebrafish. However, this does not invalidate the role of the remaining sensor pathways in other forms of genome instability. In fact, our own study supports the notion that the rescue afforded by the loss of STING (e.g. male fertility) occurs only temporarily and total loss of telomere protection in crisis (M2) results in genome instability may likely trigger RIGI and MAVS, as observed in human cell culture studies (See point 1 for Referee #1). Nevertheless, our findings are corroborated by previous studies that indicate the cGAS-STING pathway is important in aging and genomic instability. Specifically, that inhibition of STING suppresses aging-associated inflammation and neurodegeneration (PMID: 37532932) and the impaired DNA repair response of SAMHD1 deficient mice activates MDA5 in a cGAS-STING-dependent manner (PMID: 36346347).

Dear Miguel,

Thank you for submitting your revised manuscript (EMBOJ-2024-117553R) to The EMBO Journal, as well for your patience with our response. Your amended study was sent back to the referees for their scientific re-evaluation, and we have now received detailed comments from both of them, which I enclose below. As you will see, the experts state that the work has been substantially enhanced by the revisions and they are broadly in favour of publication.

Thus, we are pleased to inform you that your manuscript has been accepted in principle for publication in The EMBO Journal.

We now need you to take care of a number of issues related to formatting and data presentation as detailed below, which should be addressed at re-submission.

Please contact me at any time if you have additional questions related to below points.

Thank you for giving us the chance to consider your manuscript for The EMBO Journal. I look forward to your final revision.

Again, please contact me at any time if you need any help or have further questions.

Best regards,

Daniel

>> Author Contributions: Remove the author contributions information from the manuscript text. Note that CRediT has replaced the traditional author contributions section as of now because it offers a systematic machine-readable author contributions format that allows for more effective research assessment. and use the free text boxes beneath each contributing author's name to add specific details on the author's contribution.

More information is available in our guide to authors.
<https://www.embopress.org/page/journal/14602075/authorguide>

>> Adjust the title of the 'Declaration of Competing Interests' section to 'Disclosure and Competing Interests Statement'.

>> Remove strikethrough sections from the manuscript text.

>> References: please adjust reference format to EMBO Journal format, 10 authors et al, and place References after the Discussion, before figure legends.

>>> Appendix file with ToC: the file with suppl. information should be renamed "Appendix" and uploaded as a PDF. Please rename the suppl. tables "Appendix Table S1" etc., and the suppl. figures "Appendix Figure S1". Please add a title page with table of

contents, including page numbers.

>> Figures in separate files: main and EV figures need to be uploaded as individual, high-resolution figure files.

>> Figure callouts: callout for Fig. 2 should be before callout for Fig. 3; missing callouts for Fig. 2A-C; 4E; 5A, 5E, 5J; but there are callouts for Fig. 2F,H; 4M; 6F that do not exist. All callouts should be in consecutive order.

>> Add a Reagents and Tools table to the Methods section, as a separate file using the existing template in the Guide For Authors, listing key reagents, experimental models, software and relevant equipment.

>> Data availability section: please reference the published RNAseq data sets used (SRA; PRJNA937311); add a statement 'Otherwise, no large-scale data sets amenable to public repository deposition were generated on this study.'.

>> As to our journal policies we kindly request clarification regarding overlap within the testis data presented in Figure 4A.

>> Consider additional changes and comments from our production team as indicated below:

- Figure Legends (main + EV): 1. Please note that both the figure and figure legends for figure 5j-k is missing in the manuscript. This needs to be rectified.

2. Please note that the exact p values are not provided in the legends of figures 1b-d; 2b-c; 3b-e, g; 4b, d-e; 5b-c, i; 6a-c, e, supplementary figures 2e; 3d, g; 5.

3. Please indicate the statistical test used for data analysis in the legends of figures 1a-g.

4. Please note that the box plots need to be defined in terms of minima, maxima, centre, bounds of box and whiskers, and percentile in the legends of figures 1a-d, supplementary figure 2a.

5. Please note that the error bars are not defined in the legends of figures 1e-g.

6. Please note that scale bar and its definition are missing for figures 1e; 3a; 4a; 5a, f, h.

7. Please note that the orange arrows are not defined in the legend of figure 1e. This needs to be rectified.

Referee #1:

I am happy for how the authors have addressed both mine and reviewer #2's comments.

I have no further questions or comments.

Referee #2:

The authors have replied satisfactorily in my previous comments. The manuscript can be published in its present form.

>> As to our journal policies we kindly request clarification regarding overlap within the testis data presented in Figure 4A.

We apologize for the mistake. A panel of testis with new figures replaced the previous one and we have updated the raw data.

Dear Dr Ferreira,

Thank you for submitting the revised version of your manuscript. I have now evaluated your amended manuscript and concluded that the remaining minor concerns have been sufficiently addressed.

I am thus pleased to inform you that your manuscript has been accepted for publication in the EMBO Journal.

On a different note, I would like to alert you that EMBO Press offers a format for a video-synopsis of work published with us, which essentially is a short, author-generated film explaining the core findings in hand drawings, and, as we believe, can be very useful to increase visibility of the work. Please see the following link for representative examples and their integration into the article web page:

<https://www.embopress.org/doi/full/10.15252/embo.2019103932>

Best regards,

Daniel Klimmeck

Daniel Klimmeck, PhD
Senior Editor
The EMBO Journal
EMBO
Postfach 1022-40
Meyerhofstrasse 1
D-69117 Heidelberg
contact@embojournal.org